# Nash equilibrium of attack and defense behaviors between predators and prey

**Hiroyuki Ichijo**[1]*, **Yuichiro Kawamura**[2], **Tomoya Nakamura**[3]

**1** Department of Anatomy, Faculty of Medicine, University of Toyama, Research Centre for Idling Brain Science, University of Toyama, Sugitani, Toyama, Japan, **2** School of Medicine, University of Toyama, Sugitani, Toyama, Japan, **3** Department of Anatomy, Faculty of Medicine, University of Toyama, Research Centre for Idling Brain Science, University of Toyama, Sugitani, Toyama, Japan

* ichijo@med.u-toyama.ac.jp

## Abstract

How animals process information, compute, and execute behaviors is a central question in neuroscience and computational biology. Predators attack prey by chasing or ambushing them, while prey respond with escaping or freezing. These behaviors are fundamental for survival. Uncovering functions of such behaviors requires an understanding not only of the implementation of neuronal circuits but also of the underlying algorithms and computation. However, how animals respond to predators or prey depending on whether they can detect them from a distance remains unclear. Here, we modeled and analyzed attack and defense behaviors with game theory. Using encounter probabilities to construct payoff matrices under a sensory–motor algorithm that lacked directional information, we identified the corresponding equilibrium behaviors for the agents (predators and prey). Different detection distances yielded distinct Nash equilibrium behaviors, representing a computational mechanism that can account for diverse attack and defense behaviors. The games based on interactions among multiple predators and prey were, in most cases, non-constant-sum and positive-sum games. Measured payoffs of Nash equilibrium behaviors indicated that the predators were able to increase their payoffs by attacking, and the prey were also able to increase their payoffs even in the presence of predators. These results suggest that each of the agents initiates attack and defense behaviors. Moreover, Nash equilibrium behaviors were also identified under a simpler non-sensory motor algorithm. Despite the similarity, the non-sensory motor algorithm and the sensory–motor algorithm had distinct adaptive significance. The sensory–motor algorithm produced substantially greater prey payoffs. By implementing these algorithms, agents interact in ways that give rise to payoff matrices from which various Nash equilibrium behaviors can be mathematically derived under different conditions. Furthermore, this approach offers an experimental framework for understanding behavioral evolution and suggests a possible difference in evolutionary mechanisms of attack and defense behaviors.

**Data availability statement:** All supplemental programs and source data, together with their supporting information files, have been included in the revised submission and are listed in the Supplementary Materials section for reference.

**Funding:** HI JP21K06371 Grant-in-Aid for Scientific Research(C) of Japan Society for the Promotion of Science (JSPS KAKENHI) https://www.jsps.go.jp/j-grantsinaid/index.html The funder had no role in study design, data collection and analysis, decision to publish, or preparation of the manuscript. TN JP22K07367 Grant-in-Aid for Scientific Research(C) of Japan Society for the Promotion of Science (JSPS KAKENHI) https://www.jsps.go.jp/j-grantsinaid/index.html The funder had no role in study design, data collection and analysis, decision to publish, or preparation of the manuscript.

## Author summary

Predators attack prey by chasing or ambushing, while prey respond to predator threats with defense behaviors such as escaping or freezing. These behaviors are fundamental survival strategies for predators and prey (agents). However, how animals respond to predators or prey depending on whether they can detect them from a distance remains unclear. Using individual-based models, we simulated and analyzed predator–prey interactions and computed the Nash equilibria of the resulting payoff matrices; at these equilibria, no agent can increase its payoff by a unilateral deviation. Different detection distances resulted in distinct Nash equilibrium behaviors, including chasing and ambushing by predators, escaping and freezing by prey, and their combinations. Furthermore, predators can increase their payoffs by attacking, and prey can also increase their payoffs even in the presence of predators, suggesting that attack and defense may be initiated automatically. The results indicate a computational mechanism that generates various attack and defense behaviors. Finally, we suggest that attack and defense behaviors may follow different evolutionary mechanisms. This is due to asymmetries in predator–prey interactions because predators can learn through trial and error, whereas prey cannot learn from fatal defense failures.

## Introduction

Attacking prey and defending against predators are basic survival behaviors. Attack and defense behaviors are traits in which predators capture prey animals, and prey animals evade predators, respectively. Predators find and attack their prey by chasing (i.e., actively increasing their speed) or ambushing (i.e., stopping their movement, passively waiting, and making a surprise attack) [1,2]. On the other hand, threat stimuli such as predator smells, looming shadows, and warning sounds, automatically evoke defense behaviors in prey animals, including escaping (i.e., actively increasing their speed) and freezing (i.e., stopping their movement and passively waiting for the threat to pass) [3–15]. It has been argued that predator and prey behaviors influence the spatial arrangement between them, resulting in differences in interaction probability—here defined as the encounter probability—and directly impacting predation [16].

How predators capture prey and how prey evade predators are challenges in information processing that are likely to recur frequently in predator–prey interactions. Neuronal circuits implementing attack and defense behaviors physically embody programs at the algorithmic level, allowing for the efficient computation of solutions to these information-processing problems at the computational level, as described in the three levels postulated by Marr: implementation, algorithm, and computation [8,17]. Recent studies have demonstrated that the fundamental mechanisms of defense behaviors in mice are innate and executed by genetically hard-wired neuronal circuits [3,18–21], elucidating the level of implementation of these defense behaviors. However, algorithms and computations underlying attack and defense behaviors have been overlooked.

Therefore, numerically evaluating the modifications in the encounter probabilities caused by the agents' behaviors aids in understanding the algorithms and computations of the behaviors. As proposed by Lima [22], this requires modeling the strategic behaviors of both predator and prey, allowing the mutual influence of attack and defense behaviors on their payoffs, based on the encounter probabilities, to be analyzed using game theory [22–28]. In line with this approach, previous studies have described predator–prey interactions as a two-player, noncooperative zero-sum game and analytically elucidated the outcomes [29–31]. Chen et al. (2005) modeled the predator's attack (pursuit, ambush) and prey's defense behaviors (escape, hide) [29]. Zoroa et al. (2011) formulated sit-and-wait and active predation behaviors for predators to search for food [30]. Alpern et al. (2019) focused on the search-pursuit interaction as a stochastic iterative game in which both agents move [31]. The solution to the game is defined by the Nash equilibrium [32]. The Nash equilibrium behaviors illustrate how agents engage in specific behavioral patterns and achieve stable payoffs, where neither the predators nor the prey increase their payoffs by changing their behaviors [22,31,33,34]; however, previous studies have not determined the Nash equilibrium. Determining the Nash equilibrium presents challenges due to the complexity of the involved components. For example, multiple defense behavioral processes occur when a prey animal encounters a predator. Individual prey animals sense multiple stimuli, pay attention to the spatial locations of each stimulus source, select salient stimuli, compare the competing events in the situation, make a decision, drive their motor systems under temporal and spatial regulation, evade danger, and survive [15]. This study addresses this challenge and identifies Nash equilibria for attack–defense interactions using numerical simulations.

To understand the algorithms and computations underlying attack and defense behaviors, this study does not reproduce the complexity of the behaviors but idealizes the behaviors and presents them as interactions between a single predator and a single prey animal, or between multiple predators and prey animals in an individual-based model (IBM). An IBM has been previously used to study how the behaviors of predators (learning, attack timing, and movement) and prey (random movement, vigilance, and preferred area) affect each other's habitat spatial distribution, thereby altering encounter probabilities [35]. In this study, we modeled attack and defense behaviors using an algorithm of sensory and motor abilities, where the agents move at a constant speed by default, use the distance from the opponent as the sensory input [36], and change the speed as the motor output, although the agents have no directionality regarding which direction the opponent is or in which direction they move. Based on this framework, we computed Nash equilibria in attack and defense behaviors between the predator and prey [37]. A variety of Nash equilibrium behaviors were observed depending on the sensory abilities, including chasing and ambushing in predators, escaping and freezing in prey, and behavioral switching in both agents. Moreover, Nash equilibria were also identified in a simple algorithm without sensory input (non-sensory motor algorithm). Furthermore, the non-sensory motor and sensory–motor algorithms differed in adaptive significance, resulting in distinct outcomes for the agents. Finally, we discuss possible differences in the evolutionary mechanisms of attack and defense behaviors.

## Results

### Nash equilibria for attack and defense in a single predator and a single prey game

We first examined attack and defense behaviors between a single predator and a single prey species in a grid environment (Figs 1A, S1A). Grid worlds, commonly used in artificial intelligence and reinforcement learning, are abstract environments composed of lattice-like arrangements of squares [38], and are also referred to as lattice models in the field of ecology [39]. In this setting, agents moved between adjacent squares at a constant speed (default speed = 1), sensed the opponent within a certain range, and changed their speed accordingly. The available behaviors were decreasing the speed to 0, maintaining it at 1, or increasing it to 2, corresponding to ambush, no change, and chase behaviors for predators, and freeze, no change, and escape behaviors for prey. The predators and prey were nicknamed "wolves" (w) and "sheep" (s), respectively. The variables and measurements used in the simulations are listed in Tables 1 and 2, respectively.

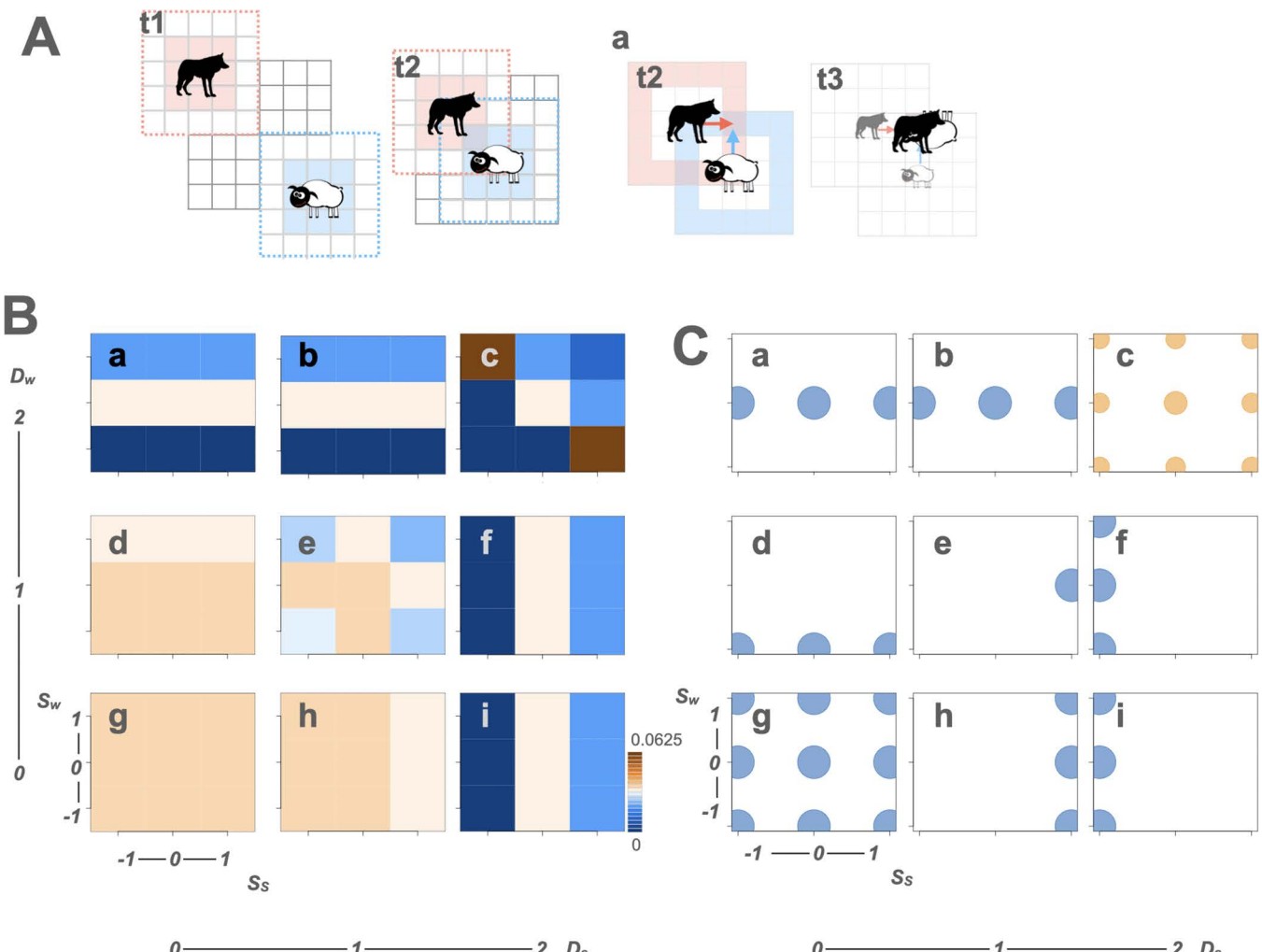

**Fig 1. Attack and defense behaviors with the sensory–motor algorithm between a single predator and a single prey.** **A** illustrates attack and defense behaviors between a single predator and a single prey (agents) in the grid world. Three time steps are shown (**t1**, **t2**, **t3**). The dotted lines illustrate their sensory abilities of detection ranges ($D_w = 2$ for the predator [wolf] and $D_s = 2$ for the prey [sheep]). At **t1**, when both agents detect no opponent, they move 1 grid per time step. The shaded grids represent possible positions they could move to at the next time step. If they detect the other at **t2**, they adjust their speeds by $S_i$ ($i = w$ for wolf and $s$ for sheep), changing their speed to $1 + S_i$. **Aa** shows a case where $S_w = S_s = 1$. The wolf chases the sheep ($1 + S_w = 2$), and the sheep escapes ($1 + S_s = 2$). If both occupy the same grid cell, as shown by the arrows in **Aa-t2**, the wolf eats the sheep at **t3**. For the illustrations of the wolf and sheep, we used open-source images from Openclipart (https://openclipart.org/detail/254708/wolf-silhouette-2; https://openclipart.org/detail/174830/sheep). **B** illustrates landscape diagrams that depict encounter probabilities based on combinations of detection distance ($D_i$). The diagrams show nine combinations, ranging from $D_s = 0$ and $D_w = 2$ (**a**) to $D_s = 2$ and $D_w = 0$ (**i**). In each landscape, the x-axis represents the sheep's speed change ($Ss$), while the y-axis represents the wolf's speed change ($Sw$). Encounter probabilities (p) associated with different combinations of $D_i$ and $S_i$ are color-coded within each landscape. High and low probabilities are indicated in dark brown and dark blue, respectively. **C** presents the Nash equilibrium behaviors on the same coordinates as the landscape diagrams. The positions of circles in each diagram indicate the prey's and the predator's Nash equilibrium speed changes ($\bar{S}_s$ and $\bar{S}_w$) on the x and y axes, respectively (**a–i**). The diameter of each circle reflects the occurrence probability of the behaviors ($\bar{o}_i$). Blue circles represent cases where Nash equilibrium behavior is uniquely determined (diameter of 1), while orange circles indicate cases where behaviors are performed probabilistically.

Each agent was characterized by its sensory ability (detection distance, $D_i$) ($i = w$ and $s$) and motor ability (magnitude of speed change, $S_i$) upon detecting an opponent. However, agents had no information regarding their own heading or the relative position of the opponent, thus operating without directional information. Different combinations of $D_i$ and $S_i$ determined

**Table 1. Variables of simulations.**

| Variables | The sensory–motor algorithm |
|---|---|
| $d$ | Sensory trait |
| $d_w$ | $d$ of predators (wolves) |
| $d_s$ | $d$ of prey (sheep) |
| $D$ | Distance to detect the opponent* |
| $D_w$ | $D$ of predators (wolves) |
| $D_s$ | $D$ of prey (sheep) |
| | **The non-sensory motor algorithm** |
| $K$ | Probability of a speed change |
| $K_w$ | $K$ of predators (wolves) |
| $K_s$ | $K$ of prey (sheep) |
| | **Common in both algorithms** |
| $S$ | Motor trait for the magnitude of speed change† |
| $S_w$ | $S$ of predators (wolves) |
| $S_s$ | $S$ of prey (sheep) |
| $c_b$ | Behavioral cost coefficient in the NetLogo world |
| $r$ | Reproduction coefficient of predators in the NetLogo world |

* The distance to detect an opponent ($D_i$) ($i = w, s$) is $d_i$ multiplied by the sensitivity factor $f$ ($D_i = d_i \times f$, where $f = 10$). †Motor traits are represented by the magnitude of speed change ($S_i$). When agents detect an opponent within radius $D_i$, they change their speed to $1 + S_i$ in the sensory–motor algorithm. The agents change their speed to $1 + S_i$ with a probability of $K_i$ in the non-sensory motor algorithm.

**Table 2. Measurements of simulations.**

| Measurements | |
|---|---|
| $p$ | Encounter probability in the grid world |
| $E$ | Encounter frequency in the NetLogo world |
| $N_w$ | Number of predators (wolves) in the NetLogo world |
| $N_s$ | Number of prey (sheep) in the NetLogo world |
| $p_w$ | Encounter probability of predators ($p_w = E/N_w$) |
| $p_s$ | Encounter probability of prey ($p_s = E/(E + N_s)$) |
| $P_w$ | Payoff of the predator in the NetLogo world ($P_w = p_w$) |
| $P_s$ | Payoff of the prey in the NetLogo world ($P_s = 1 - p_s$) |
| $P_{diff}$ | Difference between the prey and predator payoffs ($P_{diff} = P_s - P_w$) |
| $\overline{S}$ | Nash equilibrium magnitude of speed change |
| $\overline{S}_w$ | $\overline{S}$ of predators (wolves) |
| $\overline{S}_s$ | $\overline{S}$ of prey (sheep) |
| $\overline{P}$ | Nash equilibrium payoff |
| $\overline{P}_w$ | $\overline{P}$ of predators (wolves) |
| $\overline{P}_s$ | $\overline{P}$ of prey (sheep) |
| $\overline{o}$ | Occurrence probability of a Nash equilibrium behavior |
| $\overline{o}_w$ | $\overline{o}$ of predators (wolves) |
| $\overline{o}_s$ | $\overline{o}$ of prey (sheep) |

the behavioral responses of the agents. Fig 1A illustrates a representative example in which the predator chases and the prey escapes ($D_s=D_w=2$ and $S_s=S_w=1$). Initially (t1), neither agent detects the opponent, as they are outside each other's detection range (indicated by the dotted lines), and they move to adjacent shaded grids, which represent their possible next positions. At the next time step (t2), both agents detect each other, increase their speed to 2, and move to new shaded grids. If both agents are located at the same grid at the final time step (a-t2), they encounter and predation occurs (t3). To simplify the analysis, we defined "encounter" as synonymous with predation. Other scenarios, including ambush-escape ($S_w= -1$, $S_s=1$), ambush-freeze ($S_w= -1$, $S_s= -1$), and chase-freeze ($S_w=1$, $S_s= -1$) combinations, are also illustrated (S1A Fig).

Rather than simulating agent movements over multiple time steps in a finite grid, we calculated encounter probabilities ($p$) based on a single-step forward transition from initial positions where interactions could occur, under different combinations of $D_i$ and $S_i$. $p$ was calculated as (number of encounters)/(total number of cases), enumerating all initial predator–prey positions and both agents' post-first-step headings under detection-informed and uninformed behavior (S1 File). When $D_i=2$, agents could detect each other at the outer edge of the detection range, and combinations in which they were adjacent were excluded. In our setting, because encountering a predator leads directly to predation, $p$ was equivalent to the probability of predation, and $1-p$ to the probability of evading predation. Landscape diagrams illustrating $p$ values for nine combinations of $D_i$ are shown in Fig 1B. These diagrams show that, under the sensory–motor algorithm without directionality, combinations of behaviors produce diverse encounter probabilities. Comparing the colors across small squares showed that $p$ depended not only on the agent's own $S_i$ but also on the opponent's $S_i$ and both $D_i$ values.

We then regarded $p$ and $1-p$ as the predator's and prey's payoffs, respectively, and constructed 2-player payoff matrices (S1 Data, S11 Data). Because the sum of payoffs was constant ($p + (1-p) = 1$), the games were noncooperative constant-sum games. To determine game-theoretic solutions, we calculated the Nash equilibria from these payoff matrices for each $D_i$ combination. The Nash equilibrium values of $S_i$ are denoted as $\overline{S}_i$ for each agent. The $\overline{S}_i$ values for all $D_i$ combinations were plotted at corresponding coordinates on the payoff landscapes (Fig 1C, S2 Data).

We identified a single different Nash equilibrium behavior as a pure strategy, depending on the combination of the agents' sensory abilities. In particular, the prey's sensory ability ($D_s$) had a substantial effect on the Nash equilibrium behaviors of both agents. For the predator, $\overline{S}_w$ tended to depend on the prey's $D_s$. When $D_w=1$, predators adopted ambush ($\overline{S}_w= -1$) if $D_s=0$ (Fig 1Cd), and no change ($\overline{S}_w=0$) if $D_s=1$ (Fig 1Ce). When $D_s=2$, predators showed no preference, with all $\overline{S}_w$ value providing equivalent payoff (Fig 1Cf and 1 Ci). For the prey, $\overline{S}_s$ tended to depend on its own $D_s$. When $D_s=1$, prey adopted escape ($\overline{S}_s=1$) (Fig 1Ce and 1Ch). When $D_s=2$, prey adopted freeze ($\overline{S}_s= -1$) (Fig 1Cf and 1 Ci). Because freezing at long range is counterintuitive when prey have a detection-range advantage, we recomputed $p$ including adjacent configurations at $D_i=2$, which is equivalent to permitting a two-step interaction (S2 File). Under this setting, the prey's Nash equilibrium became escape when the prey's detection range exceeded the predator's (S1Cf and S1Ci Fig), indicating that the previous freeze was due to the single-step assumption, and that the more realistic escape equilibrium was identified once two-step interactions were allowed, consistent with the NetLogo model described in the next section. Similar to predators, when $D_w=2$, prey showed no clear preference across $\overline{S}_s$ values (Fig 1Ca and 1Cb).

When both agents detected each other two grids apart ($D_w=D_s=2$), they adopted mixed strategies, changing their speeds probabilistically rather than deterministically (Fig 1Cc). The occurrence probabilities ($\overline{o}_i$) for each behavioral choice were indicated by the diameters of the orange circles. Specifically, prey escaped ($\overline{S}_{s,1}=1$) with a probability of 0.363, maintained speed ($\overline{S}_{s,2}=0$) with 0.499, and froze ($\overline{S}_{s,3}= -1$) with 0.139. Similarly, predators chased ($\overline{S}_{w,1}=1$) with 0.363, maintained speed ($\overline{S}_{w,2}=0$) with 0.499, and ambushed ($\overline{S}_{w,3}= -1$) with 0.139. In contrast, when neither agent detected the opponent ($D_w=D_s=0$), no meaningful behavior change occurred and $\overline{S}_i$ could take any value (Fig 1Cg).

## Nash equilibria for attack and defense based on multiple predators and prey interactions

The game in the grid world did not incorporate agent densities, costs dependent on sensory and motor traits, or costs dependent on the resulting speed. Therefore, drawing on concepts from evolutionary game theory [40–42], we developed

an IBM that considers the effects in which multiple agents move, eat food, reproduce, pay costs, and die [35]. Using a programming language and an integrated development environment called NetLogo [43], we created a simulator to empirically solve the game (Fig 2A and 2B, S1 Movie, S3 File). This approach is called empirical game-theoretic analysis [44].

In the IBM, the agents had a sensory trait ($d_i$) and a motor trait ($S_i$), both defined without directional information. The sensory ability (detection distance $D_i$) was determined as $D_i = d_i \times f$, where $f$ is a constant coefficient representing the sensory sensitivity (factor sensitivity, $f = 10$), and $D_i$ took values of 0, 1, 2, and 3. We introduced the coefficient $f$ to normalize trait values to a common range (0–0.3) between the sensory–motor ($d_i$) and non-sensory motor algorithms ($K_i$, where $K_i$ represents the probability of speed change), enabling direct comparison. The motor traits, representing changes in speed, $S_i$, took values from −1–1 in increments of 0.2 (−1, −0.8, −0.6, −0.4, −0.2, 0, 0.2, 0.4, 0.6, 0.8, 1). The cost dependent on the sensory and motor traits was the sum of the sensory trait ($d_i$) and the absolute value (abs) of the $S_i$ multiplied by the behavioral cost coefficient ($c_b$); that is, $(d_i + abs(S_i)) \times c_b$, which corresponds to the cost associated with implementing neuronal circuit mechanisms for the behaviors. The cost dependent on the resulting speed was (1 or $1 + S_i$) multiplied by a constant, that is, the movement cost coefficient ($c_m$). Therefore, different values were subtracted from the energy of the agent at each time point when the agent changed or did not change its speed.

Combinations of $D_i$ and $S_i$ determined the agents' behaviors. By design, predator and prey agents were homogeneous and shared identical sensory and motor traits. These behavioral combinations influenced the population-level number of predator–prey encounter events per time step (encounter frequency, $E$), which in turn affected the numbers of agents ($N_i$). We assumed interactions occurred in a homogeneous, well-mixed population. Under this assumption, we approximated per-agent encounter number by the population-level $E$ as an intermediate variable and applied $E$ uniformly when evaluating payoffs. Using $E$ and $N_i$, we then computed the encounter probability at each time step.

In the NetLogo program, $E$ and $N_i$ were recorded after agent interactions. The number of predators ($N_w$) included both encountered and unencountered predators, whereas the number of prey ($N_s$) excluded prey that were removed due to predation. Thus, predator and prey encounter probabilities ($p_w$ and $p_s$) were calculated as $E/N_w$ and $E/(E + N_s)$, respectively.

In general, predation consists of both pre- and post-encounter processes [35], but in the IBM, these processes were simplified by treating the encounter itself as predation. In this framework, predators gain energy by eating prey, and prey gain energy by eating grass after evading the encounter in the IBM (S1 Table). Under the assumption of uniform interactions in a well-mixed environment, we approximated per-agent payoffs by the population-level payoffs based on the encounter probabilities. Accordingly, we defined agents' payoffs ($P_i$) as follows: predator payoffs ($P_w = p_w$), and prey payoffs ($P_s = 1 - p_s$). Because $P_i$ fluctuated over time, we averaged $P_i$ to obtain statistically stationary values after the transient in the simulation had settled. These calculations mapped behavioral combinations ($D_i$ and $S_i$) to the outcomes ($P_i$).

Fig 2C depicts the landscapes of $E$ with the values of $c_b$ (= 0.02) and the reproduction coefficient of the predators ($r = 3.6$). When $D_s = 0$ or 1, the $E$ landscape shifted smoothly with changes in $S_i$ (Fig 2Ca, 2Cb, 2Ce, 2Cf, 2 Ci, 2Cj, 2 Cm, and 2Cn). However, when $D_s = 2$ or 3, $E$ showed discontinuous drops at $S_s = 0$ (Fig 2Cc, 2 Cd, 2 Cg, 2Ch, 2Ck, 2Cl, 2Co, and 2Cp), indicating a marked decrease in $E$ when prey changed speed. The $P_s$ landscape was similar to the pattern of $E$ with inverted colors (Fig 2D); prey obtained greater payoffs when they modified their speeds (Fig 2Dc, 2Dd, 2Dg, 2Dh, 2Dk, 2Dl, 2Do, and 2Dp). By contrast, the $P_w$ landscapes shifted smoothly and continuously without discontinuous transitions (Fig 2E).

Although the IBM included many agents, we treated the entire population as a pool of potential players engaging in pairwise contests in a two-player game [45], rather than assuming that every individual was a player. Under a homogeneous, well-mixed, statistically stationary population, we used the population-level per-step encounter probability to define per-agent normalized payoffs $P_i$. Unlike the standard construction that builds a payoff matrix from an isolated single pairwise interaction, this pool-of-players approach aggregated simultaneous multi-agent interactions, thereby capturing density-dependent (population-mediated) effects in predator–prey games that the single-pair method did not reveal. The resulting bimatrix of $P_i$ was constructed as the payoff matrix of a game in which each agent selected one of 11 motor

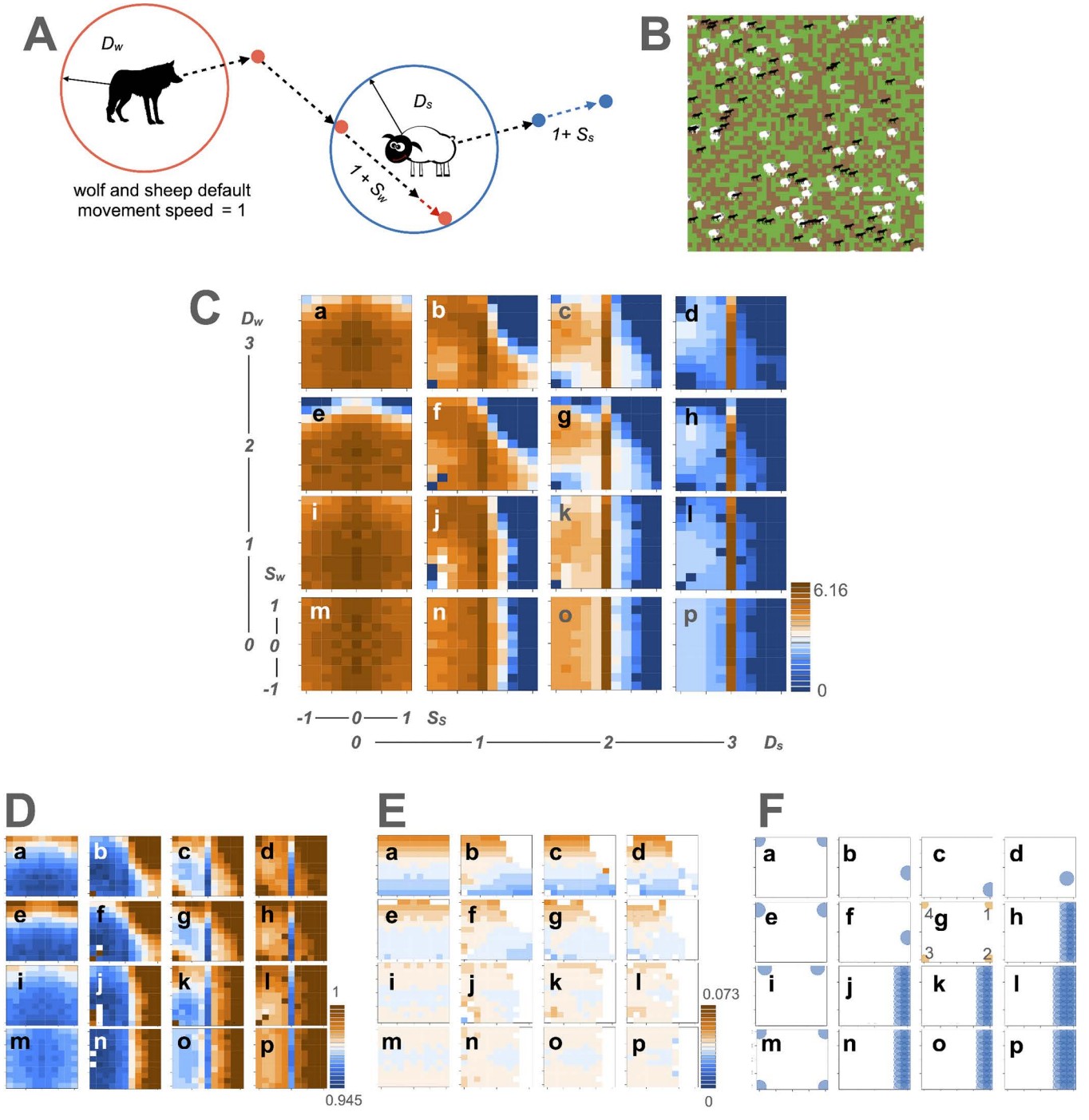

**Fig 2. Attack and defense behaviors with the sensory–motor algorithm between multiple predators and prey. A** illustrates the agents' behaviors—predators (wolves, $w$) and prey (sheep, $s$)—in the NetLogo world. Both agents move at a default speed of 1 per time step. When they detect the opponent within a detection distance ($D_i$) ($i = w$ or $s$), they change their speed in a new speed of $1 + S_i$. $D_w$ and $D_s$ are shown as red and blue circles, respectively. In this illustration ($S_w > 0$ and $S_s > 0$), they increase their speeds, corresponding to chase in predator and escape in prey, respectively. For the illustrations of the wolf and sheep, we used open-source images from Openclipart (https://openclipart.org/detail/254708/wolf-silhouette-2; https://openclipart.org/detail/174830/sheep). **B** shows the NetLogo world, where green grass grows on brown soil. Wolves (in black) move, lose energy, gain energy by eating prey, reproduce, and die if their energy reaches zero. Sheep (in white) move, eat grass to gain energy, reproduce, and die either from predation or when their energy reaches zero. **C** represents landscape diagrams showing encounter frequency ($E$) between predators and prey with the values of the behavioral cost coefficient ($c_b = 0.02$) and reproduction coefficient of the predators ($r = 3.6$). Sixteen landscapes depict different

combinations of agents' detection distance, $D_i$ (= 0, 1, 2, 3). In each landscape, the x-axis represents the prey's speed change ($S_s$) and the y-axis represents the predator's speed change ($Sw$). $E$ is color-coded. Darker brown represents higher frequencies and darker blue indicates lower frequencies. **D** and **E** show landscape diagrams for prey and predator payoffs ($P_s$ and $P_w$), respectively, following the same $D_i$ and $S_i$ combinations, and color codes as in C. **F** depicts Nash equilibrium behaviors. The circle positions represent the Nash equilibrium speed change of prey ($\overline{S}_s$) and predators ($\overline{S}_w$) on the x and y axes, respectively, in each diagram. Circle diameters reflect the occurrence probabilities ($\overline{o}_i$) of these behaviors. Blue circles indicate uniquely determined behaviors ($\overline{o}_i = 1$). Orange circles represent probabilistic behaviors ($0 < \overline{o}_i < 1$) which result in four behavioral pairs located at the corners of the landscape: chase and escape (1), ambush and escape (2), ambush and freeze (3), and chase and freeze (4) in **Fg**.

strategies ($S_i$), conditioned on its specific sensory ability ($D_i$) (S3 Data). Then, we computed Nash equilibrium speed change $\overline{S}_i$ values (S4 Data).

This analysis confirmed that Nash equilibrium behaviors depended on combinations of agents' sensory abilities, particularly the prey's. We plotted $\overline{S}_i$ values on the payoff landscape (Fig 2F). Blue circles represent pure strategies with a probability of 1 ($\overline{o}_i = 1$). For the predator, $\overline{S}_w$ tended to depend on the opponent $D_s$. When $D_s = 0$, $\overline{S}_w > 0$ (chase) was preferred (Fig 2Fa, 2Fe, and 2Fi). When $D_s \neq 0$, $\overline{S}_w < 0$ (ambush) was preferred (Fig 2Fb, 2Fc, 2Fd, and 2Ff). For the prey, $\overline{S}_s$ tended to depend on $D_s$. When $D_s \neq 0$, $\overline{S}_s > 0$ (escape) was adopted (Fig 2Fb, 2Fc, 2Fd, 2Ff, 2Fh, 2Fj, 2Fk, 2Fl, 2Fn, 2Fo, and 2Fp). When the predator did not detect prey or detected prey only at close ranges, or the prey detected predators from afar ($D_w = 0, 1, 2,$ and $D_s = 3$, or $D_w = 0, 1,$ and $D_s = 1, 2$), the predator was extinct; therefore, the $\overline{S}_w$ was not determined and assumed any value (Fig 2Fh, 2Fj, 2Fk, 2Fl, 2Fn, 2Fo, and 2Fp). When the prey did not detect the predator or change its speed ($D_w = 1, 2, 3,$ and $D_s = 0$), the prey adopted $\overline{S}_s = -1, -0.8, 0.8,$ or $1$, where high costs were required for the motor traits (Fig 2Fa, 2Fe, and 2Fi). Therefore, depending on $D_i$, the $\overline{S}_i$ value was determined; the effect of $D_s$ on $\overline{S}_i$ is noticeable.

When both agents detected each other from afar ($D_w = D_s = 2$), sets of probabilistic behavior switching constituted Nash equilibria located at the diagram corners (Fig 2Fg, orange circles). Four types of behavioral pairs identified probabilistically: chase×escape, ambush×escape, ambush×freeze, and chase×freeze. The predators chased ($\overline{S}_{w,1} = 1$) with a probability of 0.462 and ambushed ($\overline{S}_{w,2} = -1$) with 0.538. The prey escaped ($\overline{S}_{s,1} = 1$) with 0.605 and froze ($\overline{S}_{s,2} = -1$) with 0.395 (Table 3). The effects of Nash equilibrium behavioral switching were further examined in a subsequent section.

When both agents failed to detect opponents and did not change their speeds ($D_w = D_s = 0$), $\overline{S}_i$ values were located at diagram corners with large absolute values of motor traits (Fig 2Fm). This was likely because the high cost of the motor traits reduced $E$ (Fig 2Cm), indirectly increasing $P_s$ (Fig 2Dm). Furthermore, Nash equilibrium results for all $c_b$ (0, 0.001, 0.02, 0.04) and $r$ (3.2, 3.6, 4.0) are shown in S2–S5 Figs, demonstrating that distinct Nash equilibrium combinations were identified under varying conditions.

**The Nash equilibrium behavioral payoffs in attack and defense.**

To examine effects of the Nash equilibrium behaviors on agents, we assigned each agent sets of detection distance $D_i$ and Nash equilibrium speed change $\overline{S}_i$ (S5 Data) to perform the Nash equilibrium behaviors in the IBM (S4 File) and measured encounter frequency ($E$), numbers of agents ($N_i$), and the Nash equilibrium payoffs ($\overline{P}_i$) (S6 Data).

**Table 3. Occurrence probabilities of behavioral switching in multi-agent predator–prey interactions when both $D_w$ and $D_s$ are 2.**

| Agent | Behaviors | |
|---|---|---|
| Predator | $\overline{S}_{w,1} = 1$ (chase) | $\overline{S}_{w,2} = -1$ (ambush) |
| Probability | 0.462 | 0.538 |
| Prey | $\overline{S}_{s,1} = 1$ (escape) | $\overline{S}_{s,2} = -1$ (freeze) |
| Probability | 0.605 | 0.395 |

The Nash equilibrium is a mixed strategy when both $D_w$ and $D_s$ are 2; agents switch their behaviors randomly with the indicated occurrence probabilities.

In this framework, the game based on the multiple predators–prey interactions is constant-sum only under specific conditions: when the agents do not encounter each other ($E = 0$) or when the numbers of agents before interaction are equal ($N_w = E + N_s$). Under these conditions, the sum of payoffs is exactly one ($P_s + P_w = 1 − E/(E + N_s) + E/N_w = 1$). However, in general, when encounters occur ($E \neq 0$) and $N_w \neq E + N_s$, the sum deviates from one ($P_s + P_w \neq 1$). Moreover, when the total number of prey and encounters ($E + N_s$) is greater than the number of predators ($N_w$), the sum of payoffs becomes greater than one ($P_s + P_w > 1$). This positive-sum condition arises because $E/N_w − E/(E + N_s) > 0$.

Actual measurements confirmed that the game is non-constant and positive-sum in most cases. Fig 3Aa presents a scatter plot showing $N_s$ on the x-axis and $N_w$ on the y-axis, with a reference line indicating $N_s = N_w$. Most data points were located below the line, indicating that prey generally outnumber predators ($N_s > N_w$). Fig 3Ab, 3Ac, 3Ad, and 3Ae present scatter plots with $E$ on the x-axis and, respectively, $N_s$, $\overline{P}_s$, $N_w$, and $\overline{P}_w$ on the y-axis. As $E$ increased, $N_s$ and $\overline{P}_s$ decrease (Fig 3Ab and 3Ac). However, because $N_w$ was proportional to $E$ (Fig 3Ad), $\overline{P}_w$ did not change markedly as $E$ increased (Fig 3Ae). We scatter-plotted all the data of $\overline{P}_i$ for all $c_b$ and $r$ values for $\overline{P}_s$ and $\overline{P}_w$ on the x and y axes, respectively, with a reference line at $\overline{P}_s + \overline{P}_w = 1$ (Fig 3Af). Most of the points were located in the upper-right area, indicating that the sum of $\overline{P}_i$ is greater than 1 ($\overline{P}_s + \overline{P}_w > 1$), as predicted.

The finding that the game is positive-sum in most cases suggests the possibility of conditions under which both agents can obtain greater payoffs together. To explore this possibility further, we investigated the effects of detection distance $D_i$ on Nash equilibrium payoffs $\overline{P}_i$. To examine the effects of $D_s$, we compared $\overline{P}_i$ between different values of $D_s$, with a constant value for $D_w$. As $D_s$ increased, $E$ decreased (the dark blue points shifted to the left side of Fig 3B), resulting in increasing $\overline{P}_s$ (the dark blue points shifted to the upper side of Fig 3B). These results indicated that the prey's Nash equilibrium defense reduced encounters and increased prey payoffs.

To examine the effects of $D_w$, we next compared the $\overline{P}_i$ between different $D_w$, holding $D_s$ constant. The effects of $D_w$ were dependent on $D_s$. When $D_s = 0$, as $D_w$ increased, $E$ did not change markedly; however, $N_w$ decreased (the dark brown points shifted to the lower side of Fig 3Ca), resulting in increasing $\overline{P}_w$ ($P_w = E/N_w$) (the dark brown points shifted to the upper side of Fig 3Cb). Additionally, the decrease in $N_w$ led to an increase in $N_s$ (the dark brown points shifted to the right side of Fig 3Cc), which further increased $\overline{P}_s$ ($P_s = 1 − E/(E + N_s)$) (the dark brown points shifted to the right side of Fig 3Cd). Consequently, as $D_w$ increased, both $\overline{P}_w$ and $\overline{P}_s$ increased simultaneously (the dark brown points shifted to the upper right in Fig 3Cd).

These results suggest that when predators detect prey from a greater distance and attack according to the Nash equilibrium under conditions without prey defense, both predators' and prey's payoffs can increase. This outcome is due to a decrease in the number of surviving predators and an increase in prey numbers. These results suggest that prey may indirectly gain higher payoffs in the presence of predators, due to increased competition among the predators. Importantly, these findings were independent of $c_b$ and $r$. Plots of the $E$-$N_i$, $E$-$\overline{P}_i$, $N_i$, and $\overline{P}_i$ for different $D_i$ values are shown in S6 and S7 Figs. For comparison, all plots of the Nash equilibrium payoffs in the grid world are shown in S8 Fig, which illustrates a noncooperative, constant-sum game.

## Effects of Nash equilibrium behavioral switching

The Nash equilibrium behaviors exhibited either pure or mixed strategies, depending on the combinations of detection distances ($D_s$, $D_w$). Mixed strategies were observed for the combinations (1, 1), (1, 2), (1, 3), (2, 2), (2, 3), (3, 1), (3, 2), and (3, 3) for all $c_b$ and $r$ conditions. These mixed strategies resulted in probabilistic switching between behaviors; predators switched between chasing and ambushing, and prey switched between escaping and freezing (Fig 2Fg; S2Df, S2Ff, S2Fg, S3Bg, S3Dg, S3Ff, S3Fg, S3Fj, S3Fl, S4Dg, and S4Fg Figs).

To explore the functional consequences of behavioral switching, we analyzed the actual measurements of $\overline{P}_i$. Accordingly, we compared encounter frequency ($E$) with and without switching specifically for the detection distances combinations ($D_s$, $D_w$) that exhibited mixed strategies across all $c_b$ and $r$ values. Within the same ($D_s$, $D_w$) condition, runs that

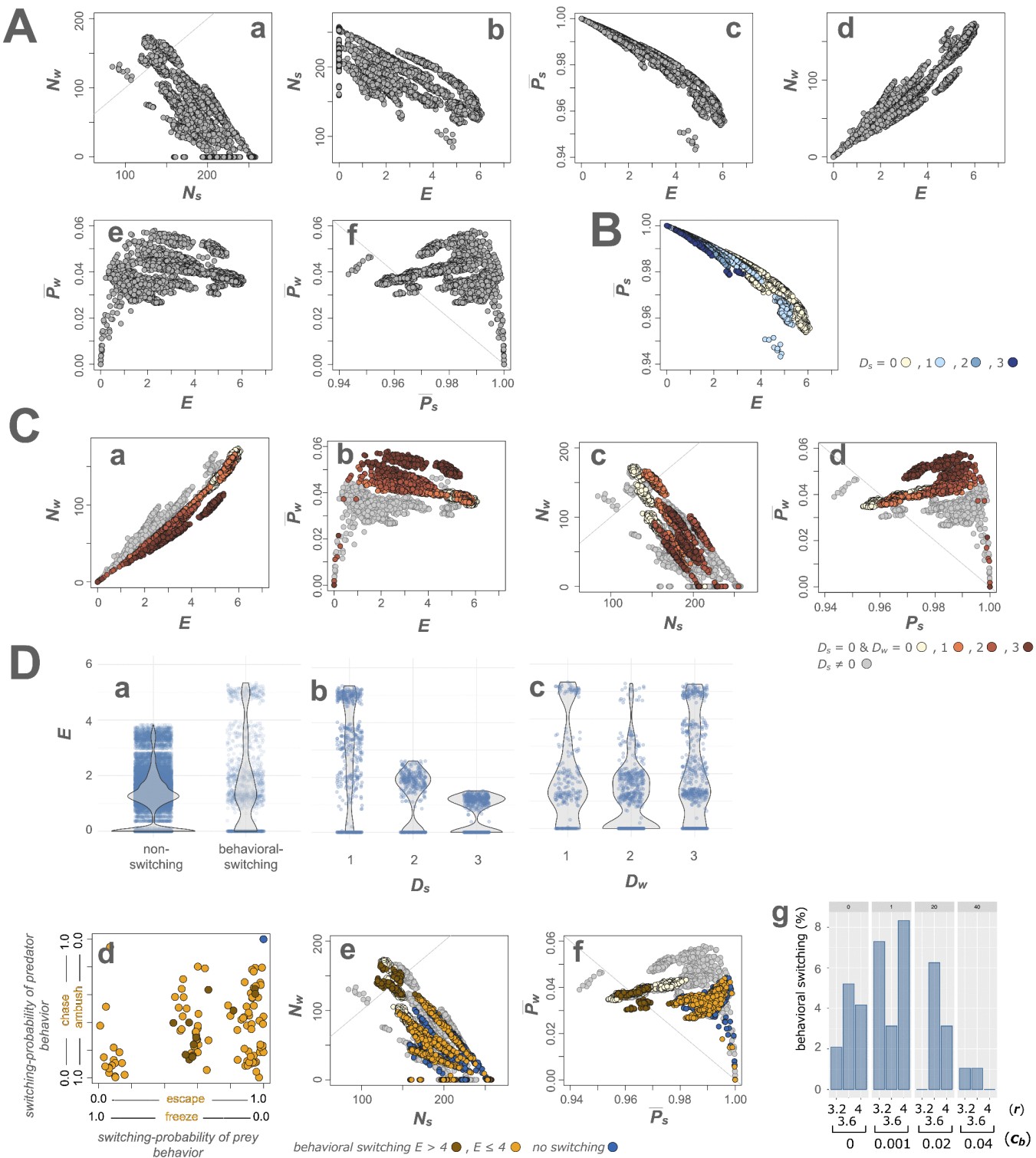

**Fig 3. Effects of Nash equilibrium attack and defense behaviors with the sensory–motor algorithm on encounters, agent numbers, and pay-offs.** **A** shows scatter plots of all data for the behavioral cost coefficients ($c_b$) and the reproduction coefficients of predators ($r$), illustrating the relationships among the number of agents ($N_i$), encounter frequency ($E$), and payoff ($\bar{P}_i$) (where $i = w$ for wolf and $s$ for sheep), based on actual measurements obtained by assigning Nash equilibrium behaviors. **Aa** shows a scatter plot of $N_s$ on the x-axis and $N_w$ on the y-axis, along the line $N_s = N_w$. **Ab**, **Ac**, **Ad**, and **Ae** show scatter plots of $E$ on the x-axis against $N_s$, $\bar{P}_s$, $N_w$, and $\bar{P}_w$ on the y-axis, respectively. **Af** shows a scatter plot of $\bar{P}_s$ on the x-axis and $\bar{P}_w$ on

the y-axis, along the line $\bar{P}_s + \bar{P}_w = 1$. **B** and **C** show the effects of detection distances $D_i$ on $E$, $N_i$, and $\bar{P}_i$. **B** shows a scatter plot of $E$ versus $\bar{P}_s$, with light blue, blue, and dark blue points indicating $D_s = 1$, 2, and 3, respectively. Beige points represent $D_s = D_w = 0$, where neither predators nor prey attack or defend. **B** indicates that $E$ decreases and $\bar{P}_s$ increases as $D_s$ increases, regardless of $D_w$ (the dark blue points shift to the upper-left side). **Ca** and **Cb** show scatter plots of $E$ on the x-axis and $N_w$ in **Ca** or $\bar{P}_w$ in **Cb** on the y-axis. **Cc** and **Cd** show scatter plots of $N_s$ versus $N_w$ and $\bar{P}_s$ versus $\bar{P}_w$, respectively. In **Ca–d**, orange, brown, and dark brown points represent $D_w = 1$, 2, and 3, respectively, where $D_s = 0$. **Ca** indicates that $N_w$ decreases as $D_w$ increases (the dark brown points shift to the lower side). **Cb** indicates that $\bar{P}_w$ increases as $D_w$ increases (the dark brown points shift to the upper side). **Cc** indicates that $N_s$ increases and $N_w$ decreases as $D_w$ increases (the dark brown points shift to the lower-right side). **Cd** indicates that $\bar{P}_s$ and $\bar{P}_w$ increase as $D_w$ increases (the dark brown points shift to the upper-right side). **D** shows the effects of behavioral switching. **Da** shows that within the same $(D_s, D_w)$ conditions, runs that exhibited behavioral switching showed higher encounter frequencies ($E > 4$) than runs without switching. **Db** and **Dc** show that the large $E$ values depend on $D_s$, with $E > 4$ when $D_s = 1$ (**Db**), but not on $D_w$ (**Dc**). **Dd, De,** and **Df** show scatter plots of behavioral switching in which point colors indicate behavioral switching with $E > 4$ in dark orange, behavioral switching with $E \leq 4$ in orange, no behavioral switching in blue within the same $(D_s, D_w)$ conditions, and other conditions without behavioral switching in gray. **Dd** shows a scatter plot of occurrence probabilities $\bar{o}_s$ versus $\bar{o}_w$ in which $\bar{o}_s$ located on the right side of the x-axis indicate a higher probability of escape and a lower probability of freeze in the prey's behavioral switching. Similarly, $\bar{o}_w$ located on the upper side of the y-axis indicate a higher probability of chase and a lower probability of ambush in the predators' behavioral switching. $\bar{o}_i$ of behavioral switching with $E > 4$ are distributed near the center (dark orange), suggesting that the choice between the two behaviors is nearly random. The agents exhibit no clear preference. Regardless of the behavioral types, the blue point representing a pure strategy (i.e., no behavioral switching) is located in the upper right of the diagram, indicating a probability of 1. **De** and **Df** show scatter plots of $N_s$ versus $N_w$ and $\bar{P}_s$ versus $\bar{P}_w$, respectively. Dark brown points shift to the upper-left side of **De**, indicating that behavioral switching with $E > 4$ led to increased $N_w$ and decreased $N_s$. Dark brown points shift to the left side of **Df**, indicating that behavioral switching with $E > 4$ decreased $\bar{P}_s$. **Dg** shows a bar plot illustrating the effect of $c_b$ on switching behavior, with percentages of switching for different values of $c_b$ and $r$. Nash equilibrium behavior switching frequently occurs when $c_b$ is small.

exhibited behavioral switching showed higher encounter frequencies ($E > 4$) than runs without switching (Fig 3Da). This increase in $E$ was particularly observed when the prey's detection distance ($D_s$) was small ($D_s = 1$) (Fig 3Db), whereas it was largely independent of the predators' detection distance ($D_w$) (Fig 3Dc).

Next, we classified behaviors into three groups: behavioral switching with $E > 4$ (dark orange), behavioral switching with $E \leq 4$ (orange), and no switching (blue), and then compared the occurrence probabilities ($\bar{o}_i$), agent numbers ($N_i$), and Nash equilibrium payoffs ($\bar{P}_i$) across these groups. In the scatter plots of $\bar{o}_i$ (Fig 3Dd), all switching behaviors were broadly distributed whereas those with $E > 4$, in particular, were centered. This indicates that the switching probabilities between the two behaviors were near 0.5, suggesting a near-random choice between the two behaviors, with no clear preference. Such near-random switching likely contributed to the unpredictability of agent behaviors.

Behavioral switching with $E > 4$ led to increased predator numbers ($N_w$) and decreased prey numbers ($N_s$) (Fig 3De). As a consequence, prey payoffs ($\bar{P}_s$) decreased, whereas predator payoffs ($\bar{P}_w$) remained relatively stable (Fig 3Df). Thus, behavioral switching increased encounter frequency and benefited predators, suggesting an adaptive strategy favoring predators.

Finally, we examined the conditions promoting switching. Increased $D_i$ tended to systematically promote mixed strategies (Figs 1Cc and 2Fg, and S2–4 and S9 Figs). Moreover, Nash equilibrium behavior switching frequently occurred at small values of $c_b$ (Fig 3Dg). In addition, we observed that only one of the agents exhibited probabilistic switching while the other did not, that is, partial behavioral switching (S9 Fig).

## Nash equilibrium speed changes in the non-sensory motor algorithm

To investigate how sensory input influences equilibrium strategies and payoffs, we tested a simple non-sensory motor algorithm. In this algorithm, agents change their speed stochastically without sensory information. This stochastic behavior was defined by two parameters: $K_i$, the probability of speed change, and $S_i$, the magnitude of speed change (Fig 4A). Using a NetLogo program (S5 File), we identified Nash equilibria under this algorithm. Across parameter combinations, $K_i$ and $S_i$ produced diverse encounter frequencies (Fig 4B), determined agent payoffs (Fig 4C and 4D), and led to Nash equilibrium speed changes ($\bar{S}_i$) (Fig 4E) ($c_b = 0.02$ and $r = 3.6$) (S7 and S8 Datas).

Analysis indicated that $\bar{S}_i$ values depended on combinations of agents' speed change probabilities $K_i$; particularly those of the prey ($K_s$). For predators, positive or negative $\bar{S}_w$ settled irrespective of $K_w$ but depended on $K_s$. When $K_s = 0$, 0.1, or

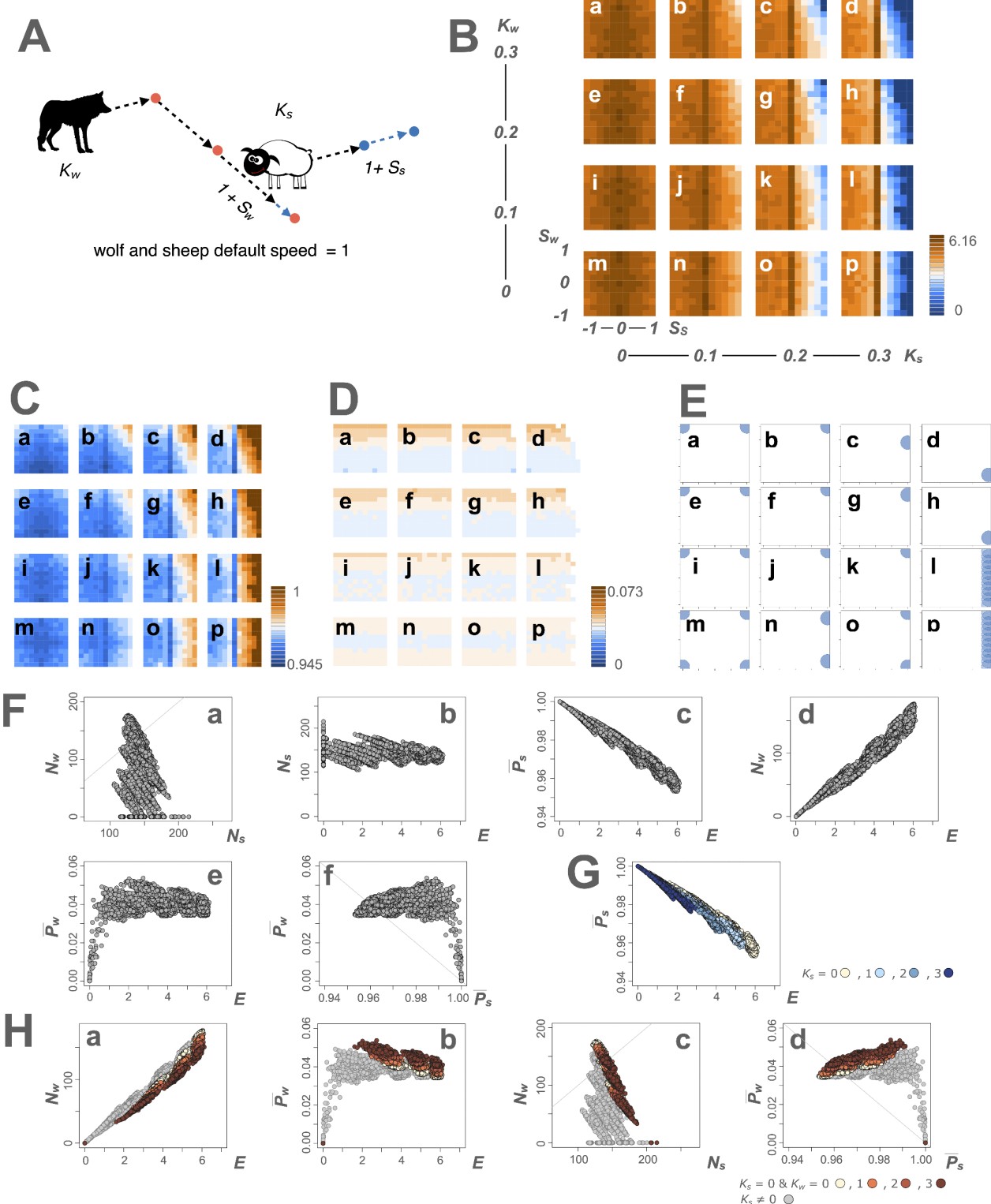

**Fig 4. Effects of Nash equilibrium speed changes with the non-sensory motor algorithm on encounters, agent numbers, and payoffs. A** illustrates that stochastic speed changes are defined by the probabilities ($K_i$) and magnitudes ($S_i$) of speed change ($i = w$ for wolf and $s$ for sheep). The agent's life history in this model is the same as the sensory–motor algorithm model, including movement, feeding, energy gain/loss, reproduction, and

death. For the illustrations of the wolf and sheep, we used open-source images from Openclipart (https://openclipart.org/detail/254708/wolf-silhouette-2; https://openclipart.org/detail/174830/sheep). **B** depicts landscape diagrams of encounter frequency ($E$) in the non-sensory motor algorithm. The landscapes represent combinations of $K_i$ (= 0, 0.1, 0.2, and 0.3). For example, **a** shows $K(0, 0.3)$, indicating $K_s = 0$ and $K_w = 0.3$, while **b** to **p** represents different combinations of $K_i$ values. $E$ is color-coded. Darker brown represents higher frequencies and darker blue indicates lower frequencies. **C** and **D** show the landscape diagrams of the prey's and the predator's payoffs ($P_s$ and $P_w$), respectively, following the same $K_i$ and $S_i$ combinations as in B. $P_j$ values are color-coded. **E** displays Nash equilibrium speed change ($\overline{S}_i$), with circles on the x and y axes indicating $\overline{S}_s$ and $\overline{S}_w$ at the same coordinates as the payoff landscape. **F** shows scatter plots of all data for the behavioral cost coefficients ($c_b$) and the reproduction coefficients of predators ($r$), illustrating the relationships among the number of agents ($N_i$), encounter frequency ($E$), and Nash equilibrium payoff ($\overline{P}_i$), based on actual measurements obtained by assigning Nash equilibrium speed changes. **Fa** shows $N_s$ on the x-axis and $N_w$ on the y-axis along the line $N_s = N_w$. **Fb**, **Fc**, **Fd**, and **Fe** plot $E$ on the x-axis against $N_s$, $\overline{P}_s$, $N_w$, and $\overline{P}_w$ on the y-axis, respectively. **Ff** plots $\overline{P}_s$ on the x-axis and $\overline{P}_w$ on the y-axis along the line $\overline{P}_s + \overline{P}_w = 1$. **G** and **H** represent the effects of $K_i$ on $E$, $N_i$, and $\overline{P}_i$. **G** shows a scatter plot of $E$ on the x-axis and $\overline{P}_s$ on the y-axis, with light blue, blue, and dark blue points representing $K_s = 0.1$, 0.2, and 0.3, respectively. Beige points indicate $K_s = K_w = 0$. **G** indicates that $E$ decreases and $\overline{P}_s$ increases as $K_s$ increases (the dark blue points shift to the upper-left side). **Ha** represents a scatter plot of $E$ on the x-axis and $N_w$ on the y-axis, with orange, brown, and dark brown points representing $K_w = 0.1$, 0.2, and 0.3, respectively, where $K_s = 0$. **Ha** indicates that $N_w$ decreases as $K_w$ increases (the dark brown points shift to the lower side). **Hb**, **Hc**, and **Hd** represent scatter plots of $E$ versus $\overline{P}_w$, $N_s$ versus $N_w$, and $\overline{P}_s$ versus $\overline{P}_w$, where $K_s = 0$. **Hb** indicates that $\overline{P}_w$ increases as $K_w$ increases (the dark brown points shift to the upper side). **Hc** indicates that $N_s$ increases and $N_w$ decreases as $K_w$ increases (the dark brown points shift to the lower-right side). **Hd** indicates that $\overline{P}_s$ and $\overline{P}_w$ increase as $K_w$ increases (the dark brown points shift to the upper-right side).

0.2, predators increased speed (such as chasing) (Fig 4Ea, 4Eb, 4Ec, 4Ee, 4Ef, 4Eg, 4Ei, 4Ej, and 4Ek). When $K_s = 0.3$, predators decreased speed (such as ambushing) (Fig 4Ed and 4Eh). For prey, positive $\overline{S}_s$ settled irrespective of $K_i$, increasing the speed (such as escaping) (Fig 4Eb, 4Ec, 4Ed, 4Ef, 4Eg, 4Eh, 4Ej, 4Ek, 4En, and 4Eo). When $K_w = 0$, 0.1, and $K_s = 0.3$, predators went extinct; therefore, $\overline{S}_w$ was not determined and did not have a single value (Fig 4El and 4Ep). When $K_w = K_s = 0$, $\overline{S}_i$ had large absolute values of motor traits, as in the sensory–motor algorithm (Fig 4Em). Thus, the prey's $K_s$ had a substantial effect on the determination of Nash equilibrium strategies. All landscapes of the payoff difference between the prey and the predator and the Nash equilibria with the non-sensory motor algorithm for all $c_b$ and $r$ are presented in S10–S13 Figs, showing that distinct Nash equilibrium combinations are identified under varying conditions.

Next, we assigned agent sets of $K_i$ and $\overline{S}_i$ (S9 Data) to perform the Nash equilibrium speed changes (S6 File), and measured encounter frequency $E$, number of agents $N_i$, and Nash equilibrium payoff $\overline{P}_i$ (S10 Data). Actual measurements showed outcomes similar to those in the sensory–motor algorithm. The game was typically a non-constant and positive-sum game. Prey's Nash equilibrium speed changes increased their payoffs by reducing encounters. Predators' Nash equilibrium speed changes also increased payoffs for both predators and prey when the prey did not change speed. As shown in Fig 4Fa, prey number ($N_s$) was usually larger than predator number ($N_w$). As $E$ increased, $N_s$ remained constant (Fig 4Fb), while $\overline{P}_s$ decreased (Fig 4Fc). However, as $N_w$ was proportional to $E$ (Fig 4Fd), $\overline{P}_w$ remained relatively constant (Fig 4Fe). The scatter-plot of $\overline{P}_s$ and $\overline{P}_w$ indicated that most points were located in the upper-right area ($\overline{P}_s + \overline{P}_w > 1$), confirming a positive-sum tendency (Fig 4Ff).

To further examine the effects of $K_i$, we compared $\overline{P}_i$ between different $K_i$ conditions. First, holding $K_w$ constant, we varied $K_s$. As $K_s$ increased, $E$ decreased, increasing $\overline{P}_s$ (the dark blue points shifted to the upper-left side of Fig 4G). Next, holding $K_s$ constant, we varied $K_w$. The effect of $K_w$ depended on $K_s$. When $K_s = 0$, as $K_w$ increased, $E$ did not change considerably but $N_w$ decreased (the dark brown points shifted to the lower side of Fig 4Ha), increasing $\overline{P}_w$ (the dark brown points shifted to the upper side of Fig 4Hb). By reducing $N_w$, $N_s$ increased (the dark brown points shifted to the right side of Fig 4Hc), increasing $\overline{P}_s$ (the dark brown points shifted to the right side of Fig 4Hd). Consequently, both $\overline{P}_w$ and $\overline{P}_s$ increased as $K_w$ increased (the dark brown points shifted to the upper-right side of Fig 4Hd). The results indicate that the prey may increase their payoff in the presence of predators. These findings were independent of $c_b$ and $r$. All plots of the $E$-$N_i$, $E$-$\overline{P}_i$, $N_i$, and $\overline{P}_i$ for different $K_i$ are shown in S14 and S15 Figs.

Despite the similarities, we found substantial differences between the non-sensory motor and sensory–motor algorithms. Compared with the non-sensory motor algorithm, the sensory–motor algorithm decreased encounter frequency ($E$) and predator number ($N_w$) (Fig 5A and 5B), while increasing prey number ($N_s$) and prey payoffs ($\overline{P}_s$) (Fig 5C and 5E).

PLOS Computational Biology

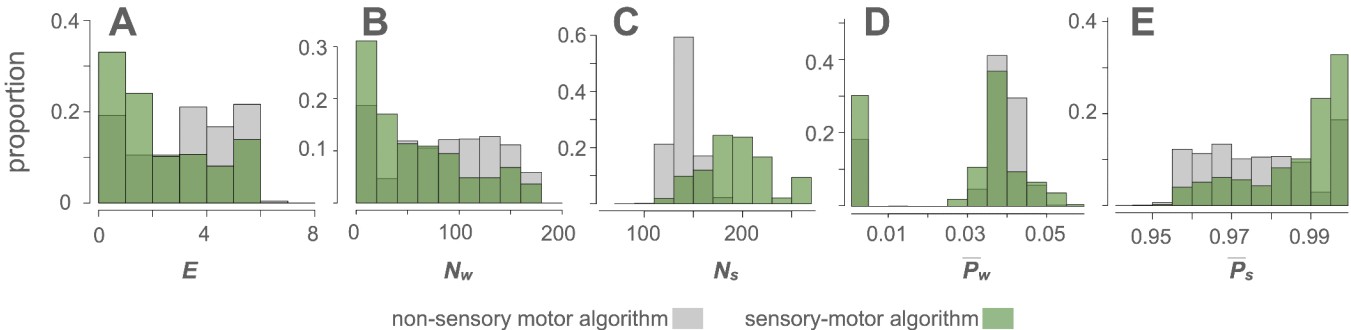

**Fig 5. Comparison between the non-sensory motor and sensory–motor algorithms.** Outcomes of the non-sensory motor algorithm and sensory–motor algorithm are compared using histograms. **A**, **B**, **C**, **D**, and **E** display the proportion of instances in histograms of encounter frequency (**E**), the number of wolves (**Nw**), the number of sheep (**Ns**), the predator Nash equilibrium payoff ($\overline{P}_w$), and the prey Nash equilibrium payoff ($\overline{P}_s$), respectively. The values for the non-sensory motor algorithm are depicted in gray, whereas those for the sensory–motor algorithm are in green. Compared to the non-sensory motor algorithm, the sensory–motor algorithm decreases $E$ (**A**) and $N_w$ (**B**) while increasing $N_s$ (**C**). The sensory–motor algorithm leads to a larger proportion of cases with $\overline{P}_w = 0$, indicating predator extinctions (**D**), and an increase in $\overline{P}_s$ (**E**).

Additionally, the sensory–motor algorithm resulted in numerous instances of $\overline{P}_w = 0$, indicating predator extinctions (Fig 5D). These results indicate that the sensory–motor algorithm results in asymmetrical effects between agents, leading to an advantage for the prey. This suggests that the sensory–motor algorithm may represent an adaptive advantage for the prey.

## Discussion

We developed a game-theoretic framework to analyze predator–prey interactions and identified Nash equilibria in payoff matrices derived from encounter probabilities in both the sensory–motor and non-sensory motor algorithms. First, we showed that, in the sensory–motor algorithm without considering directionality between a single predator and a single prey, the Nash equilibrium behaviors depended on combinations of the agents' sensory abilities, particularly those of the prey, and that the games were constant-sum and noncooperative (S8 Fig), consistent with previous studies [30,31,46].

Next, we extended this framework to multi-agent interactions. We adopted a "pool-of-players" representation: the many-agent IBM was treated as a population from which pairwise contests were drawn, and per-agent payoffs were defined from the stationary population-level encounter probability [45]. This approach differs from the conventional single-pair construction by design. It clearly mapped IBM outcomes to a two-player game and also revealed density-dependent effects that arose from simultaneous interactions. The results demonstrated a variety of Nash equilibrium behaviors, including chasing, ambushing, escaping, and freezing (Fig 2), depending on the detection distances ($D_i$), behavioral cost coefficients ($c_b$), and the reproduction coefficients of predators ($r$). In contrast to the games between a single predator and a single prey, the games in the multi-agent environment were non-constant-sum and positive-sum in most cases (Fig 3Af). When predators detected prey from a greater distance and attacked according to the Nash equilibrium, both predator and prey payoffs increased (Fig 3Cd). This outcome occurred under conditions without prey defense and was attributed to increased competition among predators (Fig 3B and 3C, S6 and S7 Figs). These results suggest that prey may obtain higher payoffs even in the presence of predators. In behavioral combinations where both agents' payoffs increase together, the predator has an incentive to attack. When the prey defends against a predator's attack, the prey's payoff increases; thus, the prey has an incentive to defend itself. These findings suggest that each of the agents initiates attack and defense behaviors.

In the sensory–motor algorithm, when detection distance was long (large $D_i$) or behavioral costs were small ($c_b = 0.001$), we observed frequent behavioral switching (Figs 1Cc, 2Fg, and 3Dg; S2–S5 and S9). Mechanistically, large $D_i$ provided both agents with earlier detection and more available headings before encounter, and the small $c_b$ reduced the weight of

the cost term. Both factors increased the influence of behavior on whether attack or defense succeeds and increased the structural complexity of the payoff landscape. The indifference condition means that a player's expected payoffs from its two behaviors are equal (given the opponent's mixture). Greater structural complexity increases the number of $S_i$ pairs that satisfy this condition. When both players satisfy the indifference condition, the best-response curves—the relationship between the opponent's behavior and the player's payoff-maximizing behavior probability—meet at probabilities strictly between 0 and 1 for both players. In that case, no pure behavior is a mutual best response, so both players must randomize between the two behaviors, yielding a mixed Nash equilibrium [47]. Consistent with this mechanism, the payoff landscapes exhibited greater structural complexity (additional bends and local extrema; Figs 1Bc and 2D–E, S1Bc and S2–S5). Together, these yielded more mixed equilibria and more frequent switching (Figs 1Cc and 2Fg, S2–S5 and S9). Importantly, the overall payoff space, that is, the spread of ($P_s$, $P_w$) pairs, was not substantially wider even in conditions where behavioral switching occurs (S16 Fig).

Behavioral switching by predators between chasing and ambushing, and by prey between escaping and freezing, resulted in stable equilibria at different combinations of the occurrence probabilities ($\bar{o}_i$ values) (Fig 3Dd). This switching increased encounter frequencies and predator number while reducing the prey payoff, providing an adaptive advantage for the predator (Fig 3D). These results suggest that implementing algorithms controlling the occurrence probability of behavioral switching in neuronal mechanisms is reasonable [48,49].

The observed changes in agent payoffs—both agents' payoffs increase through attacks without defense, and the prey payoff decreases through behavioral switching—do not appear in a single predator–prey setting (S8 Fig), indicating that they arise from density-dependent, population-mediated feedbacks specific to the multi-agent setting (i.e., multiple-agent effects).

Furthermore, we identified Nash equilibria under a non-sensory motor algorithm driven solely by movement variability (Fig 4). Games under this algorithm were typically non-constant-sum and positive-sum, similar to those using the sensory–motor algorithm. Any variation in the agent's movement mechanism, such as in neuronal circuitry [50–52] or even in simpler devices, can introduce stochastic speed fluctuations, thus generating movement variability that can be represented within a non-sensory motor algorithm. This suggests that movement variability could be an inherent mechanism contributing to equilibrium in predator–prey interactions.

We further examined how the presence of sensory input influences outcomes. Compared to the non-sensory motor algorithm, the sensory–motor algorithm was more adaptive for prey. The prey's Nash equilibrium payoffs were greater under the sensory–motor algorithm than under the non-sensory motor algorithm (Fig 5C and 5E), whereas predators' payoffs showed no such advantage (Fig 5B and 5D). This pattern reflects a structural asymmetry inherent in the IBM. Under the sensory–motor algorithm, both the predator and the prey adjust their movement timing and heading based on sensory information. The prey gains a marked advantage because any heading other than toward the predator promotes escape, whereas the predator's viable headings are mostly constrained to moving toward the target. Therefore, at the Nash equilibrium, the prey's encounter probability, $p_s$, is reduced, resulting in a higher prey payoff, $1 - p_s$, with no corresponding increase in the predator's payoff (Fig 5C and 5E vs. 5B and 5D). By contrast, under the non-sensory motor algorithm, the two agents move without cues, so encounters are governed by chance. The prey cannot detect the predator in advance and avert encounters, and thus cannot fully exploit the wider range of available headings. Therefore, the sensory–motor algorithm was more beneficial to prey than the non-sensory motor algorithm. Furthermore, among the behaviors generated by the sensory–motor algorithm, behavioral switching particularly benefited predators (Fig 3D), potentially allowing them to counter prey strategies. These functional and adaptive differences raise questions regarding their phylogenetic relationships.

We now turn to the evolutionary implications of attack and defense behaviors, which are directly subject to selection through predation [53]. We used Nash equilibria as short-term reference points for the game; agents were not presumed to know or compute the full payoff matrix. In real predator–prey settings, predators can learn from experience (successful

and failed attacks) and thereby estimate local expected payoffs using sensory–motor algorithms implemented by neural circuits, without access to all payoffs. Such learning can adjust behavior toward the short-term equilibrium predictions under stationary conditions, yielding flexibility on short time scales. We suggest that predators with neuronal mechanisms for learning, rather than directly executing innate attack behaviors, may be favored by natural selection [54–59].

In contrast, individual prey cannot reliably learn from fatal defensive failures; hence, defense is best analyzed within an evolutionary framework. The evolutionary status of these behaviors depends on whether the resident strategy resists invasion by rare mutants. In evolutionary game theory, an evolutionarily stable strategy (ESS) is a resident strategy that cannot be invaded by any sufficiently similar rare mutant under fitness-proportionate reproduction [40,60]. Because our Nash equilibrium results were derived from single interactions, this analysis did not test rare-mutant invasion; therefore, a Nash equilibrium may or may not coincide with an ESS. A formal assessment would require an Adaptive Dynamics analysis [61] by assuming a homogeneous, well-mixed, statistically stationary resident prey population playing the Nash equilibrium strategy, introducing a rare mutant, and evaluating whether it increases or declines when introduced at very low frequency. If every sufficiently similar rare mutant fails to increase, the Nash equilibrium is an ESS; otherwise, it is not.

Although it remains unknown whether the Nash equilibrium defense strategies identified here are evolutionarily stable, the corresponding behaviors—such as escaping and freezing—are often innate or developmentally specified. In mice, for example, escaping and freezing can be elicited by genetically hard-wired circuits with minimal learning [4,6–9,11,12,14,15]. The asymmetry—predators can adjust their attack behavior through experience, whereas prey cannot learn from fatal defense failures—suggests that the mechanisms differ: predators rely on strategy learning, whereas prey defenses are largely innate and genetically specified, implying distinct evolutionary bases for attack and defense [62].

A theorem by Nash (1950) guarantees the mathematical existence of Nash equilibria in any finite game [32]. This study identified the specific Nash equilibrium strategies underlying attack–defense behaviors in predator–prey interactions and clarified their behavioral significance and evolutionary implications. However, several limitations remain. First, the algorithmic effects of sensory–motor integration remain unclear and require further numerical evaluation. Second, it remains unknown how different Nash equilibrium behaviors are determined by the underlying structure of the sensory–motor algorithm, with or without directionality. Third, population dynamics were not incorporated, as Nash equilibria were computed based on average payoffs; further research may require ecological approaches and methodologies. Fourth, although our results demonstrate that movement variability alone is sufficient for Nash equilibria to exist, it remains an open question whether stochastic speed changes themselves have evolved as prototypical mechanisms that incidentally alter payoffs. Fifth, future studies could explore whether neuronal circuits mediate probabilistic switching between behaviors. Investigating this question may provide further insights into the evolution of behavioral flexibility. Finally, experimental elucidation of the evolutionary mechanisms underlying attack and defense behaviors remains a fundamental challenge. We acknowledge the absence of an ESS analysis as a significant limitation of this study. An Adaptive Dynamics analysis (e.g., via pairwise invasibility plots) would clarify whether the Nash equilibrium defense strategies identified here are evolutionarily stable. This will be a high priority for future work. Addressing these questions will help to deepen our understanding of the evolutionary mechanisms of attack and defense behaviors. Our game-theoretic framework, based on sensory and motor algorithm modeling, provides a foundation for understanding the evolution of attack and defense behaviors.

## Methods

### Formulation of attack and defense behaviors with a sensory–motor algorithm

In this study, we defined attack and defense behaviors as traits by which a predator captures prey and a prey individual evades predators, respectively. Furthermore, we defined chase–ambush as the attack behavior of predators; chase is used to detect prey and increase speed, whereas ambush is used to detect prey and reduce speed [63,64]. We defined escape–freeze as the defense behavior of the prey; escape is used to detect predators and increase speed, whereas

freeze is used to detect predators and reduce speed [15,19]. The models consisted of two types of agents: predators and prey (Figs 1A and 2A). We modeled the attack and defense behaviors with an algorithm using the distance from the opposing agent as sensory input [36] to change the speed as the motor output, but did not include sensory and motor directionalities. Agents had no directionality regarding the opponent's position or their own movement direction. The formulation was based on Edelaar's method for representing phenotypic plasticity [65]. The agents had a sensory ability, a distance to detect the opponent ($D_i$) ($i = w$ and $s$, as predator and prey, respectively). $D_i$ was the sensory trait ($d_i$) multiplied by a constant coefficient of the sensory trait (factor sensitivity, $f$), that is, $D_i = d_i \times f$, where $f = 10$. We distinguished between the latent trait $d_i$ (dimensionless sensory investment), which entered the cost function, and its spatial realization $D_i$ (detection distance), which was used to compute encounter probabilities. Information-processing costs were assessed using $d_i$ rather than $D_i$ to avoid conflating environmental units with internal investment and to maintain numerical stability across algorithms. The factor $f$ ensured cost comparability across algorithms and helped maintain simulation stability. This scaling allowed us to apply the same cost function to the sensory–motor ($d_i$) and non-sensory motor ($K_i$, the probability of speed change) formulations, placing them on a common cost basis and enabling a fair comparison. The bound $D_i \leq 3$ was a computational convenience, not a biological limit on the trait. The agents had a motor ability, a magnitude of speed change ($S_i$) when they detected their opponent. The predator had the sensory ability $D_w$ ($0 \leq D_w$) and the motor ability $S_w$ ($-1 \leq S_w \leq 1$) for its behavior, where $w$ indicates the predator (wolf). The predator moved in two-dimensional space at a speed of 1 and changed its speed to $1 + S_w$ when it found prey within a radius of $D_w$. When $S_w$ was greater than 0 ($0 < S_w$), the predator's speed increased (chase). When $S_w$ was less than 0 ($S_w < 0$), the predator's speed decreased (ambush). Without a prey individual, it moved at a speed of 1. Similarly, the prey had $D_s$ ($0 \leq D_s$) and $S_s$ ($-1 \leq S_s \leq 1$), where $s$ indicates the prey (sheep). The prey moving at speed 1 changed its speed to $1 + S_s$ when it detected a predator within radius $D_s$. At $0 < S_s$, the prey's speed increased (escape). At $S_s < 0$, the prey's speed decreased (freeze). Without a predator, it moved at a speed of 1. In addition to excluding sensory and motor directionalities, we did not consider the distance in the radius (far or close) or the sensory modality detecting the opponent [15].

**Landscape diagrams of encounter probabilities for a single predator and a single prey with the sensory–motor algorithm**

We created a grid world model to investigate the attack and defense behaviors of a predator and prey in a discrete space (Fig 1A). A single predator and a single prey were located on a two-dimensional grid (Fig 1A-t1) and moved one grid per time step (Fig 1A-t2). An agent detected the opponent in the $D_i$ range (Moore neighborhood) and changed its speed to $1 + S_i$ (Figs 1Aa-t2, S1A). $D_i$ assumed the values of 2, 1, or 0 (an agent detected the opponent two grids apart (= 2), detected the opponent in adjacent grids (= 1), or did not detect the opponent (= 0)). $S_i$ assumed the values 1, 0, or −1 (an agent changed its speed to 2 (= 1 + 1), left the speed unchanged at 1 (= 1 + 0), or changed the speed to 0 (= 1 − 1)). When the predator and prey occupied the same grid, the prey was eaten; thus, the prey died (Fig 1Aa-t3, S1A Fig). We examined encounters between agents within a single time step. Because the maximum value of $D_i$ was 2, we used a 5 × 5 grid in the x–y plane with integer coordinates, where x and y ranged from 0 to 4, inclusive. A predator was fixed at coordinate (2, 2), and the prey was placed at all possible other coordinates. When $D_i = 2$, agents could detect each other at the outer edge, and combinations in which they were adjacent were excluded. Encounter probability ($p$) was calculated in R [66] (S1 File) as (number of encounters)/(total number of cases), where "total cases" included all combinations of initial predator–prey positions and both agents' possible headings after the first step, for both detection-informed and uninformed behaviors. This procedure evaluated a single-step forward transition from all initial configurations where interaction could occur, given ($D_i$, $S_i$), rather than multi-step simulations on a finite grid. For each $D_i$, $p$ was plotted against $S_s$ and $S_w$ (Fig 1B), and the resulting Nash equilibrium behaviors are shown in Fig 1C.

When $D_s = 2$, the prey's Nash equilibrium strategy became freeze ($\overline{S}_s = -1$). It seemed unrealistic for the prey to freeze when the prey had a greater detection distance than the predator (Fig 1Cf and 1 Ci, $D_s = 2$ and $D_w = 1$ or 0). This arose

from a simplifying assumption. We recalculated $p$ including combinations in which the agents were adjacent when $D_i = 2$ (S2 File). This program corresponds to calculating encounter probabilities for a two-step interaction. Under these conditions, the prey's Nash equilibrium strategy became "escape" when it had a greater detection distance than the predator (S1Cf and S1Ci Fig). These results indicated that the "freeze" strategy observed at long range arose because only a single-step interaction was considered. When a two-step interaction was considered, the more realistic "escape" strategy was the Nash equilibrium, consistent with results from the NetLogo model.

The grid world does not include the effects of agent density or the costs associated with their behavior.

**Payoff landscapes derived from encounter probabilities in multi-agent predator–prey interactions using the sensory–motor algorithm**

We constructed a model based on modifications to the Wolf Sheep Predation model from the NetLogo Model Library [43] (Fig 2A, S3 File). The model was composed of agents (predators and prey) and patches that represent the environment. The agents moved and interacted in a continuous two-dimensional space. They could occupy any real-valued (x, y) coordinates, ensuring smooth movement. The patches served as a discrete spatial structure for managing environmental states, such as grass regrowth, but did not constrain the agents' movement. The agents were not restricted to moving from patch to patch. The agent's movement direction changes at each time step, randomly within a forward-facing 90-degree range (Fig 2B, and S1 Movie). These specifications are consistent with those used in the original Wolf Sheep Predation model provided in the NetLogo Model Library. The predators moved continuously, gained energy by consuming prey when they were located in the same patch, reproduced, lost energy as they paid costs, and died when their energy reached zero. Similarly, the prey moved continuously, gained energy by consuming grass, reproduced, lost energy as they paid costs, and died when their energy reached zero or when they were located in the same patch as a predator.

We considered three types of costs: basal metabolism, cost dependent on sensory and motor traits, and cost dependent on the resulting speed. Basal metabolism remained constant ($c_e$). The cost dependent on sensory and motor traits was the sum of the sensory trait ($d_i$) and the absolute value (abs) of the motor trait ($S_i$) multiplied by the behavioral cost coefficient ($c_b$); thus, it was calculated as follows.

$$(d_i + \text{abs}(S_i)) \times c_b$$

This corresponded to the cost associated with implementing neuronal circuit mechanisms for these behaviors. The cost of the resulting speed was calculated by multiplying the speed (1 or $1 + S_i$) by a constant, the movement cost coefficient ($c_m$), as follows.

$$(1 \ or \ 1 + S_i) \times c_m$$

Thus, in the simulation, different cost values were subtracted from the energy of the agent at each time point when the agent changed or did not change its speed. When the speed of the agent changed to $1 + S_i$, the following value was subtracted.

$$c_e + (d_i + \text{abs}(S_i)) \times c_b + (1 + S_i) \times c_m$$

The following value was subtracted when the agent did not change its speed.

$$c_e + (d_i + \text{abs}(S_i)) \times c_b + 1 \times c_m$$

Predators and prey were predetermined to perform a single behavior in the IBMs. The sensory ability (detection distance $D_i$) was $d_i$ multiplied by a constant, the coefficient of the sensory trait (factor sensitivity, $f$), that is, $D_i = d_i \times f$, where $f = 10$; $D_i$ had 4 different values (0, 1, 2, and 3). This design made the framework readily extensible to future cross-modality comparisons without redefining the cost function or losing comparability. By keeping cost on $d_i$ and relating spatial range via $D_i = d_i \times f$, the scale factor $f$ can absorb modality-specific (e.g., olfaction, vision, audition, mechanosensation) or medium-specific transduction efficiencies (e.g., transparency of air or water). In practice, using $D_i$ directly in the cost over-penalized large detection radii and made some runs unstable; the $d_i$-based scaling with $f$ avoided this while preserving the trait-to-range mapping. The value of $f$ was determined from preliminary simulation experiments.

The motor ability was equal to speed change $S_i$, which considered 11 different values ranging from −1–1 at intervals of 0.2. The combinations of $D_i$ and $S_i$ defined 44 different behaviors as their strategies of the agents. The interactions between these behavioral strategies resulted in 44 × 44 (= 1,936) unique strategy combinations. The following parameters were set as constants: energy gained by a predator eating prey, energy gained by prey from eating grass, reproduction coefficient of prey, time for grass to grow, $c_e$, $c_m$, and $f$. The following variables were used: $d_i$, $D_i$, $S_i$, $c_b$, and $r$. Values of the constants and variables used in the simulations with the sensory–motor algorithm are listed in S1 and S2 Tables, respectively.

An individual agent's behavior was defined by its detection distance ($D_i$) and speed change ($S_i$). For each parameter combination, predator and prey agents shared identical sensory and motor traits (homogeneous population) and were assumed to interact in a well-mixed environment (random pairing; no spatial structure). Under these assumptions, we treated the number of predator–prey encounter events (encounter frequency, $E$) and agents' number ($N_i$) at each time step in the population-level as intermediate variables. Both $E$ and $N_i$ varied over time as emergent properties.

Simulations were performed for all predator–prey behavior pairs using nlrx [67], which controls NetLogo in R. When each simulation was run from the beginning up to 10,000 time steps and the interaction between predators and prey reached a steady state, the following data were collected from 10,000–11,000 time steps: time step, $E$, number of wolves ($N_w$), and number of sheep ($N_s$) at each time point. As $E$, $N_w$, and $N_s$ were reported at the end of the "to go" procedure after the agents interacted in the NetLogo program, $N_w$ included the number of wolves both with an encounter and no encounter; on the other hand, $N_s$ did not include the number of sheep that died of predation with an encounter but included only with no encounter. Therefore, encounter probabilities of wolves ($p_w$) and sheep ($p_s$) were defined as $E/N_w$ and $E/(E+N_s)$, respectively. The formula appeared asymmetrical, but the encounter itself is symmetrical, with contributions from a sheep and a wolf.

Although predation typically involves both pre-encounter and post-encounter processes, our model intentionally idealized behavioral effects by focusing solely on the pre-encounter phase, and thus excluded the latter. In the IBM, energy was updated event-by-event: predators gained the constant wolf-gain-from-food only when they consumed a prey individual during an encounter, whereas prey gained sheep-gain-from-food from grazing in steps without a predator encounter. These constants were used solely for event-driven energy accounting in the IBM; their values were calibrated in preliminary simulations and are listed in S1 Table. For the payoff-matrix analysis used to identify Nash equilibria, we defined normalized payoffs. Accordingly, the payoff of wolves ($P_w$) was defined as the probability of encountering prey ($P_w = p_w$), and the payoff of sheep ($P_s$) as the probability of evading predators ($P_s = 1 - p_s$). $P_s$, $P_w$, and differences in payoffs ($P_{diff} = P_s - P_w$) were calculated at each time step. Under the assumption of uniform interactions in a well-mixed environment, we approximated per-agent payoffs by the population-level payoffs based on the encounter probabilities. These values fluctuated periodically with periods shorter than 1,000 time steps. We averaged $E$, $P_i$, and $P_{diff}$ over time steps 10,000–11,000; these averages represented post-transient, statistically stationary values.

We examined the effects of $c_b$ (0, 0.001, 0.02, and 0.04) and $r$ (1.4, 1.8, 2.8, 3.2, 3.6, and 4.0) on $E$, $P_w$, $P_s$, and $P_{diff}$. For each combination of $D_i$, 121 simulations were run, in which each engaged in 11 different behaviors (11 $S_w$ and 11 $S_s$). $E$, $P_w$, $P_s$, and $P_{diff}$ were placed on the diagrams, with $S_s$ and $S_w$ on the x and y axes, respectively. Sixteen landscape diagrams were depicted for the combinations of $D_i$. The experiments were repeated six times.

## Nash equilibrium behaviors in the sensory–motor algorithm

We studied the functions of these behaviors using game theory [31]. Attack and defense behaviors were considered as predator and prey strategies, respectively, and consisted of a strategic game between the two players [23]. In the grid world, encounter probability ($p$) was the payoff for the predator and $1-p$ was the payoff for the prey. The bimatrices with $p$ and $1-p$ were the payoff matrices of constant-sum games. The $S$ values of the Nash equilibrium ($\overline{S}_i$) were calculated under combinations of $D_i$ by using Game Theory Explorer (GTE) (https://cgi.csc.liv.ac.uk/~rahul/bimatrix_solver/)[32,68]. A script was written in Python to automate the calculations using GTE. $\overline{S}_i$ were shown at the same coordinates as those in the payoff landscape diagrams.

The NetLogo IBM included many agents. For game-theoretic analysis, we treated the population as a pool of potential players engaging in pairwise contests [45]. We defined normalized per-agent payoffs as $P_w$ and $P_s$. This differed from the conventional method of constructing payoff matrices from single isolated pairwise interactions and was intended to aggregate simultaneous multi-agent interactions, thus capturing density-dependent, population-mediated effects that were absent in single-pair settings. The resulting bimatrix of $P_i$, derived from these behaviors, served as the payoff matrix of the game, in which each agent selected one of the eleven motor strategies ($S_i$) based on its specific sensory ability ($D_i$). This framework allowed for a systematic analysis of the strategic interactions between the predator and prey within the model [69,70]. The calculations of $\overline{S}_i$ using the GTE [68] were automated using a script written in Python. $\overline{S}_i$ was shown for the combinations of $D_i$ at the same coordinates as the payoff landscape diagrams.

To examine effects of the Nash equilibrium behaviors on agents, we assigned agent sets of $D_i$ and $\overline{S}_i$ (S5 Data) to perform the Nash equilibrium behaviors (S4 File) and measured $E$, $N_i$, and Nash equilibrium payoffs ($\overline{P}_i$) (S6 Data). The experiment was repeated ten times. We examined the effects of Nash equilibrium behaviors on $E$, $N_i$, $\overline{P}_i$, switching probability, and percentages of behavioral switching by drawing scatter, violin, and bar plots.

In this study, we did not use statistical significance tests on $p$-values to interpret the simulation model results. Statistical methods are considered valuable for data with limited sample sizes, but not for simulation data with virtually unlimited sample sizes [71].

## Formulation of agents' speed changes with a non-sensory motor algorithm and Nash equilibrium in game theory

The agents' movements were formulated using a non-sensory motor algorithm. The agents set the probabilities ($K_i$) and magnitudes ($S_i$) of speed change of their stochastic speed changes (Fig 4A). The predator moving with speed 1 changed its speed to $1+S_w$ ($-1 \leq S_w \leq 1$) with the probability $K_w$ ($0 \leq K_w \leq 1$). When $S_w$ was greater than 0 ($0 < S_w$), the speed of the predator increased, corresponding to that of the predators' chasing. When $S_w$ was less than 0 ($S_w < 0$), the speed decreased, corresponding to that of the predators' ambushing. With a probability of $1-K_w$, the predator moved at a speed of 1. The prey moving with speed 1 changed its speed to $1+S_s$ ($-1 \leq S_s \leq 1$) with the probability $K_s$ ($0 \leq K_s \leq 1$). When $S_s$ was greater than 0 ($0 < S_s$), the prey's speed increased, corresponding to that of the prey's escaping. When $S_s$ was less than 0 ($S_s < 0$), the speed decreased, corresponding to that of the prey's freezing. With a probability of $1-K_s$, the prey moved at a speed of 1.

We constructed a model with the non-sensory motor algorithm in the IBM using NetLogo (S5 File). The life history of the agents was the same as that in the sensory–motor algorithm model, consisting of movement, feeding, energy gain or loss, reproduction, and death. The "costs" comprised basal metabolism, the cost of probability and motor traits, and the cost of the resulting speed. Basal metabolism and the cost of the resulting speed were the same as those in the sensory–motor algorithm model. The cost of the probability and motor traits was defined as the sum of $K_i$ and the absolute value of the motor trait (abs($S_i$)) multiplied by $c_b$, which corresponds to the cost associated with implementing neuronal circuit mechanisms for their stochastic speed change, as follows:

$$(K_i + \text{abs}(S_i)) \times c_b$$

The following value was subtracted when the agent changed its speed to $1 + S_i$.

$$c_e + (K_i + abs(S_i)) \times c_b + (1 + S_i) \times c_m$$

Conversely, the following value was subtracted when the agent did not change its speed.

$$c_e + (K_i + abs(S_i)) \times c_b + 1 \times c_m$$

The probability of speed change ($K_i$) had four values (0, 0.1, 0.2, and 0.3). Except for $K_i$, the other parameters (values of the constants and variables) were the same as those of the sensory–motor algorithm model. The values of the constants and variables used in the simulations with the non-sensory motor algorithm are listed in S3 and S4 Tables, respectively. In addition, the measurements and data collection were the same as those for the sensory–motor algorithm model. This experiment was repeated 6 times. The supplementary source data show the payoff matrices for all the experiments (S7 Data).

 The Nash equilibrium speed changes were computed in the same manner as those in the sensory–motor algorithm using the GTE with the script written in Python. The results for all the experiments are shown in the Supplementary Source Data (S8 Data). The results of the non-sensory motor algorithm model were depicted in the same manner as those of the sensory–motor algorithm model; $E$, $P_i$, and $\overline{S}_i$ were shown in the landscape diagrams.

 To examine the effects of the Nash equilibrium speed changes on agents, we assigned agent sets of $K_i$ and $\overline{S}_i$ (S9 Data) to perform the Nash equilibrium speed changes (S6 File) and measured $E$, $N_i$, and $\overline{P}_i$ (S10 Data). The experiment was repeated ten times. We examined the effects of the Nash equilibrium speed changes on $E$, $N_i$, and $\overline{P}_i$ in the non-sensory motor algorithm model by drawing scatter plots. Furthermore, we compared the effects of the non-sensory motor and the sensory–motor algorithms on $E$, $N_i$, and $\overline{P}_i$ by drawing histograms.

## Supporting information

**S1 Fig. Attack and defense behaviors between a single predator and a single prey using the sensory–motor algorithm. A** illustrates the attack and defense behaviors between a single predator and a single prey (agents) in a grid world. Three time steps are shown (**t1, t2, t3**). The dotted lines represent their detection ranges ($D_w = 2$ for the predator [wolf] and $D_s = 2$ for the prey [sheep]). At **t1**, neither agent detects the opponent and each moves one grid per time step. The shaded grids indicate the possible positions they may occupy at the next time step. At **t2,** if they detect each other, both adjust their speeds by $S_i$ ($i = w$ for wolf and $s$ for sheep), changing their speed to $1 + S_i$. **Aa** shows a case where $S_w = S_s = 1$. The wolf chases the sheep ($1 + S_w = 2$), and the sheep escapes ($1 + S_s = 2$). If both occupy the same grid cell, as indicated by the arrows in **Aa-t2**, the wolf eats the sheep at **t3**. **Ab** shows a case where $S_w = -1$ and $S_s = 1$. The wolf ambushes ($1 + S_w = 0$), and the sheep escapes ($1 + S_s = 2$). If both occupy the same grid cell, as shown by the arrow in **Ab-t2**, the wolf eats the sheep at **t3**. **Ac** shows a case where $S_w = -1$ and $S_s = -1$. The wolf ambushes ($1 + S_w = 0$), and the sheep freezes ($1 + S_s = 0$). Both remain in place and do not encounter. **Ad** shows a case where $S_w = 1$ and $S_s = -1$. The wolf chases ($1 + S_w = 2$), and the sheep freezes ($1 + S_s = 0$). If both occupy the same grid cell, as shown by the arrow in **Ad-t2**, the wolf eats the sheep at **t3**. For the illustrations of the wolf and sheep, we used open-source images from Openclipart (https://openclipart.org/detail/254708/wolf-silhouette-2; https://openclipart.org/detail/174830/sheep). **B** illustrates land-scape diagrams that depict encounter probabilities based on combinations of detection distance ($D_i$). For this S1B Fig, we calculated encounter probabilities including combinations in which the agents were adjacent when $D_i = 2$ (S2 File). This program corresponds to calculating encounter probabilities for a two-step interaction. The diagrams show nine combinations, ranging from $D_s = 0$ and $D_w = 2$ (**a**) to $D_s = 2$ and $D_w = 0$ (**i**). In each landscape, the x-axis represents the sheep's speed change ($S_s$), while the y-axis represents the wolf's speed change ($S_w$). Encounter probabilities ($p$) associated with different

combinations of $D_i$ and $S_i$ are color-coded within each landscape. High and low probabilities are indicated in dark brown and dark blue, respectively. **C** presents the Nash equilibrium behaviors under the two-step interaction. The positions of circles in each diagram indicate the prey's and the predator's Nash equilibrium speed changes ($\bar{S}_s$ and $\bar{S}_w$) on the x and y axes, respectively (**a–i**), on the same coordinates as the landscape diagrams. Under the two-step interaction, the prey's Nash equilibrium strategy became "escape" when the prey had a greater detection distance than the predator (**f**, **i**). Circle diameters reflect the occurrence probabilities ($\bar{o}_i$) of these behaviors. Blue circles indicate uniquely determined behaviors ($\bar{o}_i = 1$). Orange circles represent probabilistic behaviors ($0 < \bar{o}_i < 1$).
(TIF)

**S2 Fig. Landscapes of payoff differences ($P_{diff}$) and Nash equilibrium speed changes ($\bar{S}_i$) with the sensory–motor algorithm in the absence of the behavioral cost coefficient ($c_b = 0$). A** shows the landscapes of $P_{diff}$ under the reproduction coefficient of predators ($r = 3.2$). Sixteen landscapes depict different combinations of agents' detection distance, $D_i$ (= 0, 1, 2, 3) (**a–p**). In each landscape, the x-axis represents the prey's speed change ($S_s$) and the y-axis represents the predator's speed change ($S_w$). $P_{diff}$ values are color-coded: darker brown indicates higher payoff differences favoring prey, and darker blue indicates lower payoff differences (or advantage to predators). **B** depicts Nash equilibrium speed changes in sixteen landscapes under different combinations of agents' detection distance under $r = 3.2$ (**a–p**). Circle positions represent the Nash equilibrium speed changes of prey ($\bar{S}_s$) and predators ($\bar{S}_w$) on the x-axis and y-axis, respectively, at the same coordinates as the $P_{diff}$ landscapes. Circle diameters reflect the occurrence probabilities ($\bar{o}_i$) of these behaviors. Blue circles indicate uniquely determined behaviors ($\bar{o}_i = 1$). Orange circles represent probabilistic behaviors ($0 < \bar{o}_i < 1$). **C** and **D** show those under $r = 3.6$. **E** and **F** show those under $r = 4.0$.
(TIF)

**S3 Fig. Landscapes of payoff differences ($P_{diff}$) and Nash equilibrium speed changes ($\bar{S}_i$) with the sensory–motor algorithm under the small behavioral cost coefficient ($c_b = 0.001$). A** and **B** show the landscapes of $P_{diff}$ and Nash equilibrium speed changes, respectively, under the reproduction coefficient of predators ($r = 3.2$). **C** and **D** show those under $r = 3.6$. **E** and **F** show those under $r = 4.0$.
(TIF)

**S4 Fig. Landscapes of payoff differences ($P_{diff}$) and Nash equilibrium speed changes ($\bar{S}_i$) with the sensory–motor algorithm under the middle behavioral cost coefficient ($c_b = 0.02$). A** and **B** show the landscapes of $P_{diff}$ and Nash equilibrium speed changes, respectively, under the reproduction coefficient of predators ($r = 3.2$). **C** and **D** show those under $r = 3.6$. **E** and **F** show those under $r = 4.0$.
(TIF)

**S5 Fig. Landscapes of payoff differences ($P_{diff}$) and Nash equilibrium speed changes ($\bar{S}_i$) with the sensory–motor algorithm under the large behavioral cost coefficient ($c_b = 0.04$). A** and **B** show the landscapes of $P_{diff}$ and Nash equilibrium speed changes, respectively, under the reproduction coefficient of predators ($r = 3.2$). **C** and **D** show those under $r = 3.6$. **E** and **F** show those under $r = 4.0$.
(TIF)

**S6 Fig. Effects of prey's detection distance ($D_s$) on encounter frequency ($E$), agent numbers ($N_i$), and Nash equilibrium payoffs ($\bar{P}_i$), with agents adopting Nash equilibrium behaviors under the sensory–motor algorithm across all conditions ($c_b = 0, 0.001, 0.02, 0.04$, and $r = 3.2, 3.6, 4.0$). A** shows scatter plots of all data across the behavioral cost coefficients ($c_b$) and the reproduction coefficients of predators ($r$), illustrating the relationships among the encounter frequency ($E$), number of agents ($N_i$), and Nash equilibrium payoff ($\bar{P}_i$) (where $i = w$ for wolf and $s$ for sheep), based on actual measurements obtained by assigning Nash equilibrium behaviors. **Aa** shows a scatter plot of $E$ on the x-axis against $N_s$ on the y-axis,

and **Ab** shows a scatter plot of $E$ on the x-axis against $\overline{P}_s$ on the y-axis. **Ac** shows a scatter plot of $N_s$ versus $N_w$, along the line $N_s = N_w$. **Ad** shows a scatter plot of $\overline{P}_s$ versus $\overline{P}_w$, along the line $\overline{P}_s + \overline{P}_w = 1$. Beige, light blue, blue, and dark blue points represent $D_s = 0$, 1, 2, and 3, respectively, across all values of $D_w$ (0–3). **Ba–Bd** show the same scatter plots, with $D_w = 0$ highlighted in the colors and $D_w \neq 0$ in gray. **Ca–Cd** show the same scatter plots, with $D_w \neq 0$ highlighted in the colors and $D_w = 0$ in gray. Beige points represent $D_s = D_w = 0$ in **B** and **C**. Regardless of $D_w$, increasing $D_s$ reduces $E$ (dark blue points shift to the left side in **Aa**, **Ba**, and **Ca**), increases $\overline{P}_s$ (dark blue points shift to the upper-left side in **Ab**, **Bb**, and **Cb**, and dark blue points shift to the right side in **Ad**, **Bd**, and **Cd**), and reduces $N_w$ (dark blue points shift to the lower side in **Ac**, **Bc**, and **Cc**).
(TIF)

**S7 Fig. Effects of predators' detection distance ($D_w$) on encounter frequency ($E$), agent numbers ($N_i$), and Nash equilibrium payoffs ($\overline{P}_i$) with agents adopting Nash equilibrium behaviors under the sensory–motor algorithm across all conditions ($c_b = 0$, 0.001, 0.02, 0.04, and $r = 3.2$, 3.6, 4.0).** **A** shows scatter plots of all data across the behavioral cost coefficients ($c_b$) and the reproduction coefficients of predators ($r$), illustrating the relationships among the encounter frequency ($E$), number of agents ($N_i$), and Nash equilibrium payoff ($\overline{P}_i$) (where $i = w$ for wolf and $s$ for sheep), based on actual measurements obtained by assigning Nash equilibrium behaviors. **Aa** and **Ab** show scatter plots of $E$ on the x-axis against $N_w$ and $\overline{P}_w$ on the y-axis, respectively. **Ac** shows a scatter plot of $N_s$ versus $N_w$, along the line $N_s = N_w$. **Ad** shows a scatter plot of $\overline{P}_s$ versus $\overline{P}_w$, along the line $\overline{P}_s + \overline{P}_w = 1$. Beige, orange, brown, and dark brown points represent $D_w = 0$, 1, 2, and 3, respectively, across all values of $D_s$ ($D_s = 0$–3). **Ba–Bd** show the same scatter plots, with $D_s = 0$ highlighted in the colors and $D_s \neq 0$ in gray. When $D_s = 0$, as $D_w$ increases, $N_w$ decreases (dark brown points shift to the lower side in **Ba**). $\overline{P}_w$ increases (dark brown points shift to the upper side in **Bb**). $N_s$ increases and $N_w$ decreases (dark brown points shift to the lower-right side in **Bc**). $\overline{P}_s$ and $\overline{P}_w$ increase (dark brown points shift to the upper-right side in **Bd**). **Ca–Cd** show the same scatter plots, with $D_s \neq 0$ highlighted in the colors and $D_s = 0$ in gray.
(TIF)

**S8 Fig. Nash equilibrium payoffs ($\overline{P}_i$) with the sensory–motor algorithm between a single predator and a single prey.** Scatter plots show Nash equilibrium payoffs ($\overline{P}_i$) (where $i = w$ for wolf and $s$ for sheep), with $\overline{P}_s$ on the x-axis and $\overline{P}_w$ on the y-axis. **a–c** show effects of predator's detection distance ($D_w$) on the payoffs $\overline{P}_i$, while holding the prey's detection distance ($D_s$) constant ($D_s = 0$ in **a**, 1 in **b**, and 2 in **c**). Gray, orange, and brown points represent $D_w = 0$, 1, and 2, respectively. **d–f** show the effects of $D_s$ on $\overline{P}_i$ while holding $D_w$ constant ($D_w = 0$ in **d**, 1 in **e**, and 2 in **f**). Gray, light blue, and blue points represent $D_s = 0$, 1, and 2, respectively. **g** shows the combined effects of $D_w$ and $D_s$ on $\overline{P}_i$. Gray, light green, and dark green points represent $D_w = D_s = 0$, $D_w = D_s = 1$, and $D_w = D_s = 2$, respectively. All plots are on the diagonal line ($\overline{P}_s + \overline{P}_w = 1$), demonstrating that the games between a single predator and prey are constant-sum. In **b**, **c**, and **f**, the point sizes are modified to make their overlaps visible.
(TIF)

**S9 Fig. Partial behavioral switching.** At Nash equilibrium, we observed two forms of probabilistic speed change: partial behavioral switching, in which only one agent changed speed probabilistically (type 1), and behavioral switching, in which both predator and prey did so (type 2). **A** illustrates both types of switching across various behavioral cost coefficients ($c_b$) and the reproduction coefficients of predators ($r$) under a detection distance for both agents ($D_w = D_s = 2$). The upper (**a–d**), middle (**e–h**), and lower rows (**i–l**) correspond to $r = 4.0$, 3.6, and 3.2, respectively. The left (**a**, **e**, and **i**), the middle-left (**b**, **f**, and **j**), the middle-right (**c**, **g**, and **k**), and the right columns (**d**, **h**, and **l**) correspond to $c_b = 0$, 0.001, 0.02, and 0.04, respectively. Circle positions on the x- and y-axes indicate the Nash equilibrium speed changes of prey ($\overline{S}_s$) and predators ($\overline{S}_w$), respectively, aligned with the payoff landscape coordinates. Orange circles indicate the cases where the predator and prey probabilistically perform their behaviors with increasing and decreasing speeds. The diameters of the orange circles are proportional to the occurrence probabilities of the behaviors ($\overline{o}_i$) ($0 < \overline{o}_i < 1$). The prey increase or decrease their

speeds (either escape or freeze), whereas the predators do not; i.e., partial behavioral switching is observed in **e**. Both agents probabilistically increase or decrease their speeds in **a**, **b**, **c**, **f**, **g**, and **j**. Blue circles indicate the cases where $\overline{S}_i$ is uniquely determined ($\overline{o}_i = 1$) in **d**, **h**, **i**, **k**, and **l**. **B** shows percentages of behavioral switching in one of the agents (type 1, partial behavioral switching) and both agents (type 2, behavioral switching) across all combinations of $c_b$ and $r$.
(TIF)

**S10 Fig. Landscapes of payoff differences ($P_{diff}$) and Nash equilibrium speed changes ($\overline{S}_i$) with the non-sensory motor algorithm in the absence of the behavioral cost coefficient ($c_b = 0$).** **A** and **B** show the landscapes of $P_{diff}$ and Nash equilibrium speed changes, respectively, under the reproduction coefficient of predators ($r = 3.2$). **C** and **D** show those under $r = 3.6$. **E** and **F** show those under $r = 4.0$. Each panel presents the strategic outcomes for different combinations of prey and predator speed changes. $P_{diff}$ values are color-coded: darker brown indicates higher payoff differences favoring prey, and darker blue indicates lower payoff differences (or advantage to predators). Circle positions in **B**, **D**, and **F** correspond to the Nash equilibrium speed changes ($\overline{S}_s$ on the x-axis, $\overline{S}_w$ on the y-axis). Circle diameters reflect the occurrence probabilities of the behaviors ($\overline{o}_i$), with blue circles indicating unique equilibria ($\overline{o}_i = 1$).
(TIF)

**S11 Fig. Landscapes of payoff differences ($P_{diff}$) and Nash equilibrium speed changes ($\overline{S}_i$) with the non-sensory motor algorithm under the small behavioral cost coefficient ($c_b = 0.001$).** **A** and **B** show the landscapes of $P_{diff}$ and Nash equilibrium speed changes, respectively, under the reproduction coefficient of predators ($r = 3.2$). **C** and **D** show those under $r = 3.6$. **E** and **F** show those under $r = 4.0$. Circle diameters reflect the occurrence probabilities of the behaviors ($\overline{o}_i$), with blue circles indicating unique equilibria ($\overline{o}_i = 1$), and orange circles representing mixed strategies ($0 < \overline{o}_i < 1$).
(TIF)

**S12 Fig. Landscapes of payoff differences ($P_{diff}$) and Nash equilibrium speed changes ($\overline{S}_i$) with the non-sensory motor algorithm under the middle behavioral cost coefficient ($c_b = 0.02$).** **A** and **B** show the landscapes of $P_{diff}$ and Nash equilibrium speed changes, respectively, under the reproduction coefficient of predators ($r = 3.2$). **C** and **D** show those under $r = 3.6$. **E** and **F** show those under $r = 4.0$. Circle diameters reflect the occurrence probabilities of the behaviors ($\overline{o}_i$), with blue circles indicating unique equilibria ($\overline{o}_i = 1$), and orange circles representing mixed strategies ($0 < \overline{o}_i < 1$).
(TIF)

**S13 Fig. Landscapes of payoff differences ($P_{diff}$) and Nash equilibrium speed changes ($\overline{S}_i$) with the non-sensory motor algorithm under the large behavioral cost coefficient ($c_b = 0.04$).** **A** and **B** show the landscapes of $P_{diff}$ and Nash equilibrium speed changes, respectively, under the reproduction coefficient of predators ($r = 3.2$). **C** and **D** show those under $r = 3.6$. **E** and **F** show those under $r = 4.0$. Circle diameters reflect the occurrence probabilities of the behaviors ($\overline{o}_i$), with blue circles indicating unique equilibria ($\overline{o}_i = 1$), and orange circles representing mixed strategies ($0 < \overline{o}_i < 1$).
(TIF)

**S14 Fig. Effects of prey's speed change probability ($K_s$) on encounter frequency ($E$), agent numbers ($N_i$), and Nash equilibrium payoffs ($\overline{P}_i$) under the non-sensory motor algorithm across all conditions ($c_b = 0, 0.001, 0.02, 0.04$, and $r = 3.2, 3.6, 4.0$).** **A** shows scatter plots for all data of the behavioral cost coefficients ($c_b$) and the reproduction coefficients of predators ($r$), illustrating the relationships among the encounter frequency ($E$), number of agents ($N_i$), and Nash equilibrium payoff ($\overline{P}_i$) (where $i = w$ for wolf and $s$ for sheep), based on actual measurements obtained by assigning Nash equilibrium speed changes. **Aa** shows a scatter plot of $E$ on the x-axis against $N_s$ on the y-axis, and **Ab** shows a scatter plot of $E$ on the x-axis against $\overline{P}_s$ on the y-axis. **Ac** shows a scatter plot of $N_s$ versus $N_w$, along the line $N_s = N_w$. **Ad** shows a scatter plot of $\overline{P}_s$ versus $\overline{P}_w$, along the line $\overline{P}_s + \overline{P}_w = 1$. Beige, light blue, blue, and dark blue points represent $K_s = 0, 0.1, 0.2$, and $0.3$, respectively, across all values of $K_w$ ($K_w = 0$–$0.3$). **Ba**–**Bd** show the same scatter plots, with $K_w = 0$ highlighted in the colors

and $K_w \neq 0$ in gray. **Ca–Cd** show the same scatter plots, with $K_w \neq 0$ highlighted in the colors and $K_w = 0$ in gray. Beige points represent $K_s = K_w = 0$ in **B** and **C**. Regardless of $K_w$, increasing $K_s$ reduces $E$ (dark blue points shift to the left side in **Aa**, **Ba**, and **Ca**), increases $\overline{P}_s$ (dark blue points shift to the upper-left side in **Ab**, **Bb**, and **Cb**, and dark blue points shift to the right side in **Ad**, **Bd**, and **Cd**), and reduces $N_w$ (dark blue points shift to the lower side in **Ac**, **Bc**, and **Cc**).
(TIF)

**S15 Fig. Effects of predators' speed change probability ($K_w$) on encounter frequency ($E$), agent numbers ($N_i$), and Nash equilibrium payoffs ($\overline{P}_i$) under the non-sensory motor algorithm across all conditions ($c_b = 0$, 0.001, 0.02, 0.04, and $r = 3.2$, 3.6, 4.0).** **A** shows scatter plots for all data of the behavioral cost coefficients ($c_b$) and the reproduction coefficients of predators ($r$), illustrating the relationships among the encounter frequency ($E$), number of agents ($N_i$), and Nash equilibrium payoff ($\overline{P}_i$) (where $i = w$ for wolf and $s$ for sheep), based on actual measurements obtained by assigning Nash equilibrium speed changes. **Aa** and **Ab** show scatter plots of $E$ on the x-axis against $N_w$ and $\overline{P}_w$ on the y-axis, respectively. **Ac** shows a scatter plot of $N_s$ versus $N_w$, along the line $N_s = N_w$. **Ad** shows a scatter plot of $\overline{P}_s$ versus $\overline{P}_w$, along the line $\overline{P}_s + \overline{P}_w = 1$. Beige, orange, brown, and dark brown points represent, $K_w = 0$, 0.1, 0.2, and 0.3, respectively, across all values of $K_s$ ($K_s = 0$–0.3). **Ba–Bd** show the same scatter plots, with $K_s = 0$ highlighted in the colors and $K_s \neq 0$ in gray. When $K_s = 0$, as $K_w$ increases, $N_w$ decreases (dark brown points shift to the lower side in **Ba**). $\overline{P}_w$ increases (dark brown points shift to the upper side in **Bb**). $N_s$ increases and $N_w$ decreases (dark brown points shift to the lower-right side in **Bc**). $\overline{P}_s$ and $\overline{P}_w$ increase (dark brown points shift to the upper-right side in **Bd**). **Ca–Cd** show the same scatter plots, with $K_s \neq 0$ highlighted in the colors and $K_s = 0$ in gray.
(TIF)

**S16 Fig. Payoff spaces under $D_i$ combinations where behavioral switching occurred and under all other $D_i$ combinations of non-behavioral switching.** The pooled $P_s$ and $P_w$ data calculated across all $S_i$ combinations under $D_i$ combinations where behavioral switching occurred are designated as "bs" (behavioral switching) [$D_s$ and $D_w$ are (1, 1), (1, 2), (1, 3), (2, 2), (2, 3), (3, 1), (3, 2), and (3, 3) for all $cb$ and $r$], and those under all other $D_i$ combinations as "nbs" (non-behavioral switching). The full range of payoffs from all behaviors is plotted on a scatter plot with $P_s$ on the x-axis and $P_w$ on the y-axis. The data corresponding to bs are shown in dark gray, and those for nbs in light gray plots. Ranges of each region represent the payoff spaces of bs and nbs. **A** shows a scatter plot with nbs on top of bs. **B** shows bs on top of nbs. No marked difference is observed between the two payoff spaces.
(TIF)

**S1 Table. Constant values used in simulations involving multiple agents with the sensory-motor algorithm.**
(DOCX)

**S2 Table. Variable values used in simulations involving multiple agents with the sensory-motor algorithm.**
(DOCX)

**S3 Table. Constant values used in simulations involving multiple agents with the non-sensory motor algorithm.**
(DOCX)

**S4 Table. Variable values used in simulations involving multiple agents with the non-sensory motor algorithm.**
(DOCX)

**S1 Movie. NetLogo world movie.**
(MOV)

**S1 File. Grid world Program 1.txt (R script).**
(TXT)

**S2 File. Grid world Program 2.txt (R script).**
(TXT)

**S3 File. Sensory–Motor Program.nlogo (NetLogo program).**
(NLOGO)

**S4 File. Actual Measurement Sensory–Motor Program.nlogo (NetLogo program).**
(NLOGO)

**S5 File. Non-Sensory Motor Program.nlogo (NetLogo program).**
(NLOGO)

**S6 File. Actual Measurement Non-Sensory Motor Program.nlogo (NetLogo program).**
(NLOGO)

**S1 Data. Payoff matrices of grid world model.**
(XLSX)

**S2 Data. Nash equilibria Si of grid world model.**
(XLSX)

**S3 Data. Payoff matrices of sensory–motor IBM.**
(XLSX)

**S4 Data. Nash equilibria Si of sensor–motor IBM.**
(XLSX)

**S5 Data. Nash equilibrium conditions of sensory–motor IBM.**
(XLSX)

**S6 Data. Actual measurement of Nash equilibrium payoffs in sensory–motor IBM.**
(CSV)

**S7 Data. Payoff matrices of non-sensory motor IBM.**
(XLSX)

**S8 Data. Nash equilibria Si of non-Sensory motor IBM.**
(XLSX)

**S9 Data. Nash equilibrium conditions of non-sensory motor IBM.**
(XLSX)

**S10 Data. Actual measurement of Nash equilibrium payoffs in non-sensory motor IBM.**
(CSV)

**S11 Data. Source data.**
(PDF)

## Acknowledgments

We thank Dr. Masato Yamamichi for discussion and critical reading of the manuscript, Prof. Hiroshi Nishimaru, Prof. Rahul Savani, and Dr. Yuka Kusui for critical reading of the manuscript.

## Author contributions

**Conceptualization:** Hiroyuki Ichijo.

**Data curation:** Hiroyuki Ichijo, Yuichiro Kawamura, Tomoya Nakamura.

**Formal analysis:** Hiroyuki Ichijo, Yuichiro Kawamura, Tomoya Nakamura.

**Funding acquisition:** Hiroyuki Ichijo, Tomoya Nakamura.

**Investigation:** Hiroyuki Ichijo, Yuichiro Kawamura, Tomoya Nakamura.

**Methodology:** Hiroyuki Ichijo.

**Project administration:** Hiroyuki Ichijo.

**Resources:** Hiroyuki Ichijo.

**Software:** Hiroyuki Ichijo, Yuichiro Kawamura, Tomoya Nakamura.

**Supervision:** Hiroyuki Ichijo.

**Validation:** Hiroyuki Ichijo.

**Visualization:** Hiroyuki Ichijo.

**Writing – original draft:** Hiroyuki Ichijo.

**Writing – review & editing:** Hiroyuki Ichijo, Yuichiro Kawamura, Tomoya Nakamura.

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
