## [Decision Letter · Decision Letter 0]

4 Mar 2025

PCOMPBIOL-D-24-01911

Nash equilibrium of attack and defense behaviors between predators and prey

PLOS Computational Biology

Dear Dr. Ichijo,

Thank you for submitting your manuscript to PLOS Computational Biology. After careful consideration, we feel that it has merit but does not fully meet PLOS Computational Biology's publication criteria as it currently stands. Both Reviewers agree on the merit of your manuscript, but they raise important conceptual concerns about your modeling framework. Your revised submission must fully address these important issues. We invite you to submit a revised version of the manuscript that addresses the points raised during the review process.

Please submit your revised manuscript within 60 days May 04 2025 11:59PM. If you will need more time than this to complete your revisions, please reply to this message or contact the journal office at ploscompbiol@plos.org. Please include the following items when submitting your revised manuscript:

We look forward to receiving your revised manuscript.

Kind regards,

Ricardo Martinez-Garcia

Academic Editor

PLOS Computational Biology

Tobias Bollenbach

Section Editor

PLOS Computational Biology

**Journal Requirements:**

At this stage, the following Authors/Authors require contributions: Hiroyuki Ichijo, Yuichiro Kawamura, and Tomoya Nakamura. Please ensure that the full contributions of each author are acknowledged in the "Add/Edit/Remove Authors" section of our submission form.

2) Please include your title page at the beginning of your manuscript file that lists full author names and institute addresses.  This should not be uploaded as a separate file.

5) We have noticed that you have a list of Supporting Information legends (Supplementary Programs and Supplementary Source Data) in your manuscript. However, there are no corresponding files uploaded to the submission. Please upload them as separate files with the item type 'Supporting Information'.

6) We notice that your supplementary Figures, and Tables are included in the manuscript file. Please remove them and upload them with the file type 'Supporting Information'. Please ensure that each Supporting Information file has a legend listed in the manuscript after the references list.

7) Some material included in your submission may be copyrighted. According to PLOSu2019s copyright policy, authors who use figures or other material (e.g., graphics, clipart, maps) from another author or copyright holder must demonstrate or obtain permission to publish this material under the Creative Commons Attribution 4.0 International (CC BY 4.0) License used by PLOS journals. Please closely review the details of PLOSu2019s copyright requirements here: PLOS Licenses and Copyright. If you need to request permissions from a copyright holder, you may use PLOS's Copyright Content Permission form.

Potential Copyright Issues:

i) Figures 1A, 2A, 4A, and S1. Please confirm whether you drew the images / clip-art within the figure panels by hand. If you did not draw the images, please provide (a) a link to the source of the images or icons and their license / terms of use; or (b) written permission from the copyright holder to publish the images or icons under our CC BY 4.0 license. Alternatively, you may replace the images with open source alternatives. See these open source resources you may use to replace images / clip-art:

8) Please amend your detailed Financial Disclosure statement. This is published with the article. It must therefore be completed in full sentences and contain the exact wording you wish to be published.

1) State what role the funders took in the study. If the funders had no role in your study, please state: "The funders had no role in study design, data collection and analysis, decision to publish, or preparation of the manuscript.".

**Reviewers' comments:**

Reviewer's Responses to Questions

**Comments to the Authors:**

**Please note that two reviews are uploaded as attachments.**

Reviewer #1: Comments uploaded as a file named "PCOMPBIOL-D-24-01911_review.docx"

Reviewer #2: The manuscript investigates predator-prey dynamics in an agent-based model through a game-theoretic approach. Hereby it focuses on an idealized behavioral model. It starts by analyzing the interaction of a single prey and a single predator in a lattice environment and identifying Nash equilibria depending on sensory and motor capabilities of both predator on prey agents. Then it moves on to a multi-agent scenario, which is analyzed in detail. Finally, in addition to the full sensory-motor algorithm controlling the behavior, the authors also explored the dynamics of the system with agents using a simple motor algorithm independent of any sensory input.

The topic of the manuscript is certainly interesting, and of potential interest to theoretical/computational biologists. I also do acknowledge that there is a gap of systematic investigation of spatially explicit agent-based models for predator-prey dynamics and how physical and sensory constraints may affect evolutionary dynamics and optimal behavioral strategies. Some of the results are quite nice, like the identification of different prey and predator strategies depending on the sensory capabilities of both agent types, or investigation of equilibrium strategies for a pure motor algorithm, independent on the sensory input.

However the present manuscript has a number of important issues, and highly problematic parts. This issues need to be thoroughly addressed, before the paper can be at all considered for publication with PLoS Comp. Biol. In its current form it is not acceptable. Below, I address the different points in detail. In addition, I provide an annotated PDF with some additional comments (partly overlapping).

Major points:

1) The authors follow a game-theoretic approach, however the multi-agent scenario appears to resemble more models studied in spatially-explicit evolutionary games. Here, the authors seem to completely ignore existing - and relevant literature on evolutionary games and evolutionary stable strategies:

Mitchell, W. A. (2009). Multi‐behavioral strategies in a predator–prey game: an evolutionary algorithm analysis. Oikos, 118(7), 1073-1083.

or a review-type perspective article:

Lima, S. L. (2002). Putting predators back into behavioral predator–prey interactions. Trends in Ecology & Evolution, 17(2), 70-75.

2) Linked to the first point: The authors make claims about evolutionary selection of the Nash equilibrium (NE) strategies. However, when talking about evolutionary selection, the more appropriate concepts are those of biological fitness and of evolutionary stable strategy (ESS). Here it has to be highlighted that every ESS is a NE, but not every NE is an ESS. This needs to be clearly distinguished, and corresponding statements on the relevance to natural evolution should be reviewed and potentially toned down.

3) There is another conceptual issue: The multi-agent scenario, does not really align with a pure game-theoretic approach. Given that the agents reproduce and interact through predation, the system approaches stationary numbers (densities) of prey Ns and predators Nw, respectively. So Ns, Nw are emergent variables the entire system converges to, analogous to fixed-points in Lottka-Volterra dynamical models, and are not related to individual-level strategy choice, which is what game theory and Nash equilibrium (NE) are concerned with. Thus discussing Ns,Nw as part of NE appears ill defined, and may potentially lead to biologically problematic conclusions (see bvelow).

4) One core interpretation of the results does make biologically no sense. In the discussion section (line 334) it reads: "...the prey have an incentive to receive an attack...". However, individual prey have never an incentive to receive attacks. What is observed here are likely "spourious" relative benefits as a consequence of Ns and Nw being emergent variables (not fixed parameters). Furthermore, this may be also linked to the NE not being an ESS. Thus the corresponding discussion parts need a thorough revision.

5) The multiplayer game is insufficiently explained. For example, it is stated in the methods section that the model is in continuous space. However, in the screenshots it is apparent that at least the "grass" resource appears discretion, also an encounter (i.e. predation event) happens if prey and predator are in the same patch. Here, I assume "patch" refers to the lattice site.

Furthermore, it is not clear what type of movement the agents perform in the 2D multi-agent simulation. If this is continuous in space and time, then it could be either some sort of random walk or some sort of active Brownian motion.

Minor points:

- The language needs some revision. There are some suggestions/examples highlighted and annotated in the attached PDF.

- The article is also rather hard to read as the results are quite dense with a lot of variables, which make it hard to recall them. I suggest adding some redundancy by spelling out the names of different parameters at multiple locations, e.g. writing "encounter frequency E" instead of only writing "E" or similar for other variables.

**Have the authors made all data and (if applicable) computational code underlying the findings in their manuscript fully available?**

Reviewer #1: Yes

Reviewer #2: Yes

PLOS authors have the option to publish the peer review history of their article (what does this mean? ). If published, this will include your full peer review and any attached files.

**Do you want your identity to be public for this peer review?** For information about this choice, including consent withdrawal, please see our Privacy Policy .

Reviewer #1: **Yes:** Gaurav Athreya

Reviewer #2: No

**Figure resubmission:**
---

## [Decision Letter · Decision Letter 1]

22 Jul 2025

PCOMPBIOL-D-24-01911R1

Nash equilibrium of attack and defense behaviors between predators and prey

PLOS Computational Biology

Dear Dr. Ichijo,

Thank you for submitting your manuscript to PLOS Computational Biology. After careful consideration, we feel that it has merit but does not fully meet PLOS Computational Biology's publication criteria as it currently stands. Therefore, we invite you to submit a revised version of the manuscript that addresses the points raised during the review process.

Please submit your revised manuscript within 30 days Sep 21 2025 11:59PM. If you will need more time than this to complete your revisions, please reply to this message or contact the journal office at ploscompbiol@plos.org. Please include the following items when submitting your revised manuscript:

We look forward to receiving your revised manuscript.

Kind regards,

Ricardo Martinez-Garcia

Academic Editor

PLOS Computational Biology

Tobias Bollenbach

Section Editor

PLOS Computational Biology

**Additional Editor Comments :**

While both Reviewers agree that the authors have significantly improved the manuscript in this round of revision, they still have some concerns that must be fully addressed. In particular, I agree with both Reviewers that the manuscript would benefit from clearer, model-specific clarification of the relationship between Nash equilibria and Evolutionarily Stable Strategies (ESS). Additionally, some key results still lack a mechanistic explanation (e.g., an increased detection distance leads to a mixed strategy Nash equilibrium). Providing these explanations is essential to enhance the manuscript's biological relevance and thus adhere to PLOS Computational Biology publication criteria.

**Journal Requirements:**

1) Please ensure that "Supplementary Fig. legends" on page 49 line 1471 is removed from the supporting files legends.

**Reviewers' comments:**

Reviewer's Responses to Questions

**Comments to the Authors:**

**Please note that two reviews are uploaded as attachments.**

Reviewer #1: Uploaded as attachment "PCOMPBIOL-D-24-01911_review2.docx"

Reviewer #2: I attach my review as a PDF file.

**Have the authors made all data and (if applicable) computational code underlying the findings in their manuscript fully available?**

Reviewer #1: Yes

Reviewer #2: Yes

PLOS authors have the option to publish the peer review history of their article (what does this mean? ). If published, this will include your full peer review and any attached files.

**Do you want your identity to be public for this peer review?** For information about this choice, including consent withdrawal, please see our Privacy Policy .

Reviewer #1: **Yes:** Gaurav Athreya

Reviewer #2: No

**Figure resubmission:**
---

## [Decision Letter · Decision Letter 2]

31 Oct 2025

PCOMPBIOL-D-24-01911R2

Nash equilibrium of attack and defense behaviors between predators and prey

PLOS Computational Biology

Dear Dr. Ichijo,

Thank you for submitting your manuscript to PLOS Computational Biology. Both Reviewers are now supportive of publication, but have a few minor comments remaining that should be addressed before I recommend publication. I do not anticipate that such a revised submission will undergo another round of review.

Please submit your revised manuscript within 30 days Dec 31 2025 11:59PM. If you will need more time than this to complete your revisions, please reply to this message or contact the journal office at ploscompbiol@plos.org. Please include the following items when submitting your revised manuscript:

We look forward to receiving your revised manuscript.

Kind regards,

Ricardo Martinez-Garcia

Academic Editor

PLOS Computational Biology

Tobias Bollenbach

Section Editor

PLOS Computational Biology

**Reviewers' comments:**

Reviewer's Responses to Questions

**Comments to the Authors:**

Reviewer #1: Thanks to the authors for their revisions. I think they have now given sufficiently detailed arguments for their choices to be critically evaluated by a reader, and put their results in the appropriate context. I think the results are interesting and the method will push people to reflect on the limits of game theoretical arguments in complex ecological scenarios. I have only a few final suggestions (see below), but I am overall satisfied with the manuscript and happy to recommend that this paper be accepted!

On the usage of “admit a Nash equilibrium” and similar: I am sorry to be annoying and point this out yet again, but I am afraid I have not communicated my point well enough. So to prevent even longer discussion on this topic, I will try to be a bit more direct.

- on L67 and L568: it is still misleading to say that “This algorithm was sufficient to modify agents’ payoffs and yielded payoff matrices that admitted Nash equilibria” (L568). This is because all algorithms would generate some payoff matrix, and all payoff matrices admit a Nash equilibrium! Therefore, no information can be gained by saying that the algorithm being considered leads to a payoff matrix that admits a NE. Wording of the form “we identify NE..” that the authors use elsewhere is appropriate and resolves my concerns completely. The useful information in this study is to know what exactly the NE is when the simulations are analysed game-theoretically, and NOT that it exists. I suggest the authors remove the phrase “that admit Nash equilibria” on L67 and remove the sentence on L568 altogether.

- Line 576: “Given that movement variability alone can generate equilibria..”. anything will generate equilibria! rephrase. Perhaps the authors wish to differentiate between pure- and mixed-strategy equilibria in the above sentences. However I don’t think this is the case because pure- and mixed-strategy NE exist under both the sensory-motor and non-sensory-motor algorithms.

- Line 629 “This study demonstrated the mathematical existence of Nash equilibria in predator–prey interactions”. This is technically true, but Nash’s result (PNAS 1950, ref below) already guaranteed the mathematical existence of this study's Nash equilibria. Another fact that is perhaps useful to reiterate: the paper of Nash (1950) applies also to non-constant-sum games since the proof only uses continuity of the payoff functions, which is satisfied here. To borrow author phrasing, this study _identifies_ what these Nash equilibria should be in predator-prey interactions.

The above critique does not, in my opinion, at all take away from the usefulness of this study. I only bring it up again to prevent inaccurate statements from being made.

A typo that the authors might want to change: “Porf.” on Line 936

----

Nash JF. Equilibrium Points in N-Person Games. Proc Natl Acad Sci U S A. 1950 Jan;36(1):48-9. doi: 10.1073/pnas.36.1.48.

Reviewer #2: The authors addressed my points in a satisfactory way.

**Have the authors made all data and (if applicable) computational code underlying the findings in their manuscript fully available?**

Reviewer #1: Yes

Reviewer #2: Yes

PLOS authors have the option to publish the peer review history of their article (what does this mean? ). If published, this will include your full peer review and any attached files.

**Do you want your identity to be public for this peer review?** For information about this choice, including consent withdrawal, please see our Privacy Policy .

Reviewer #1: **Yes:** Gaurav Athreya

Reviewer #2: No

**Figure resubmission:**
---

## [Editor Report · Decision Letter 3]

10 Nov 2025

Dear Prof. Ichijo,

We are pleased to inform you that your manuscript 'Nash equilibrium of attack and defense behaviors between predators and prey' has been provisionally accepted for publication in PLOS Computational Biology.

Best regards,

Ricardo Martinez-Garcia

Academic Editor

PLOS Computational Biology

Tobias Bollenbach

Section Editor

PLOS Computational Biology

---

## [Editor Report · Acceptance letter]

PCOMPBIOL-D-24-01911R3

Nash equilibrium of attack and defense behaviors between predators and prey

Dear Dr Ichijo,

I am pleased to inform you that your manuscript has been formally accepted for publication in PLOS Computational Biology. Your manuscript is now with our production department and you will be notified of the publication date in due course.

With kind regards,

Zsofia Freund
